# Elevating the Tradeoff between Privacy and Utility in Zeroth-Order Vertical Federated Learning

## Abstract

Vertical Federated Learning (VFL) enables collaborative training with feature-partitioned data, yet remains vulnerable to private label leakage through gradient transmissions. In this work, we propose `DPZV`, a gradient-free VFL framework that achieves tunable differential privacy (DP) with formal performance guarantees. By leveraging zeroth-order (ZO) optimization, `DPZV` eliminates explicit gradient exposure. It further enhances security by providing provable differential privacy guarantees. Standard DP techniques like DP-SGD are difficult to apply in zeroth-order VFL due to VFL's distributed nature and the high variance incurred by vector-valued noise. `DPZV` overcomes these limitations by injecting low-variance scalar noise at the server, enabling controllable privacy with reduced memory overhead. We conduct a comprehensive theoretical analysis showing that `DPZV` attains convergence rate comparable to first order (FO) optimization methods while satisfying formal $(\epsilon, \delta)$-DP guarantees. Experiments on image and language benchmarks demonstrate that `DPZV` outperforms several baselines in terms of achieved accuracy under a wide range of privacy constraints ($\epsilon \leq 10$), thereby elevating the privacy-utility tradeoff in VFL.

## 1 Introduction

Vertical Federated Learning (VFL) has emerged as a compelling paradigm for scenarios where different institutions each hold complementary features for the same set of users. For example, hospitals may store patients' medical records while insurance companies possess demographic or financial information, and jointly training models on such partitioned data can unlock richer predictive power than any single party could achieve alone (Hardy et al., 2017; Chen et al., 2020a; Castiglia et al., 2023). VFL achieves this by maintaining local submodels at each party and coordinating training through the exchange of intermediate representations and gradients, without directly sharing raw data. However, unlike conventional Federated Learning (FL) where only model parameters are communicated, VFL's reliance on transmitting intermediate results introduces new attack surfaces. Recent studies have shown that adversaries can exploit these updates to recover sensitive attributes or even infer labels, leading to feature leakage (Jin et al., 2021; Ye et al., 2024) and label leakage (Fu et al., 2022; Zou et al., 2022). These vulnerabilities underscore the urgent need for integrating stronger, principled privacy-preserving mechanisms into the VFL framework.

To mitigate privacy risks introduced by the transmission of intermediate gradients, one promising direction is to integrate Zeroth-order (ZO) optimization (Nesterov & Spokoiny, 2017; Fang et al., 2022) into VFL (Zhang et al., 2021; Wang et al., 2024). By avoiding gradient transmission, ZO substantially reduces the risk of label leakage, offering a stronger baseline defense against label inference. (Zhang et al., 2021). Beyond privacy, ZO-VFL further offers practical advantages such as reduced backward communication and lower memory overhead. However, ZO remains vulnerable for two key reasons: (i) malicious clients can still approximate gradients from perturbation information, and (ii) adversaries may infer sensitive attributes from local model parameters. Moreover, ZO provides no built-in mechanism for adjustable privacy, limiting their flexibility in practical deployments where institutions require adaptive privacy guarantees. These limitations raise the central research question of this work:

*How can we enable an enhanced and controllable privacy mechanism in zeroth-order VFL which maintains convergence guarantees while elevating the privacy-utility tradeoff?*

**Key Challenges.** In this paper, we take a differential privacy (DP) (Dwork et al., 2006) approach to zeroth-order VFL, aiming to provide adjustable privacy levels under varying budgets. However, incorporating DP into ZO-based VFL is not a straightforward adaptation and poses several key challenges. In particular, existing strategies for achieving DP within the VFL framework typically involve injecting vector-valued noise into the forward embeddings (Chen et al., 2020a; Xie et al., 2024). While providing formal privacy guarantees, it requires adding noise with the same shape as the transmitted embeddings, which are typically high-dimensional. This high-dimensional noise introduces substantial variance amplification (Abadi et al., 2016). Moreover, ZO methods inherently produce noisy gradient estimates (Ghadimi & Lan, 2013): when combined with the additional variance introduced by DP noise, the compounded effect can lead to substantial performance degradation. This situation can be further exacerbated when the privacy budget is tight. On the other hand, conventional DP algorithms(e.g., DP-SGD(Abadi et al., 2016)) injects noise on the gradient, which is also incompatible with ZO methods, as ZO does not involve transmitting backward gradients.

**Contributions.** To address the above challenges, we propose Differentially Private Zeroth-Order VFL (`DPZV`), a novel ZO-based VFL framework that achieves a controllable differential privacy level with convergence guarantees. By injecting *scalar-valued* noise rather than high-dimensional vector noise, `DPZV` mitigates variance amplification, thereby enhancing the tradeoff between privacy and model performance in VFL. Our main contributions can be summarized as follows:

- `DPZV` is the first ZO optimization framework for vertical federated learning (VFL) that enables tunable differential privacy. By injecting calibrated scalar-valued noise instead of high-dimensional vector noise, `DPZV` provides privacy guarantees while preserving model accuracy. Moreover, the use of ZO optimization substantially reduces memory overhead, making `DPZV` particularly suitable for privacy-critical and resource-constrained environments (section 3).

- We provide a rigorous theoretical analysis establishing both convergence and privacy guarantees for `DPZV`. Specifically, we show that `DPZV` achieves a convergence rate on the same order as first-order DP-SGD, despite using only ZO estimators. This result indicates that our `DPZV` framework attains privacy without sacrificing training efficiency. Furthermore, we prove that our method satisfies $(\epsilon, \delta)$-DP, confirming its ability to provide an adjustable privacy control mechanism (sections 4 and 5).

- Through experiments on image and language benchmarks, we demonstrate that `DPZV` obtains robust convergence performance under strict privacy budgets. While other DP baselines experience a steep performance degradation as the privacy level increases, `DPZV` consistently maintains high accuracy across all tasks, underscoring an elevated tradeoff between privacy and utility (section 6).

## 2 BACKGROUND AND MOTIVATION

**System Model for VFL.** We consider a VFL framework with one server and $M$ clients. In VFL, data is vertically partitioned between clients, with each client holding different features of the data. Suppose we have a dataset $\mathcal{D}$ with $D$ samples: $\mathcal{D} = \{\xi_i | i = 1, 2, \ldots, D\}$. Each data sample $\xi_i$ can be partitioned into $M$ portions distributed throughout all clients, where the data sample on machine $m$ with ID $i$ is denoted as $\xi_{m,i}$, hence $\xi_i = [\xi_{1,i}, \xi_{2,i}, \ldots, \xi_{M,i}]^\top$. The server is numbered as machine 0 and holds the label data $\boldsymbol{Y} = \{y_i | i = 1, 2, \ldots, D\}$.

The clients and the server aim to jointly train a machine learning model parameterized by $\boldsymbol{w}$. The global model comprises local models on each party parameterized by $w_m$, with $m = 0, 1, \ldots, M$ being the ID of the machine. To protect privacy, clients do not communicate with each other regarding data or model parameters. Instead, all clients communicate directly with the server regarding their local model's outputs, which we term as local embeddings. If we denote the local embedding of client $m$ as $h_{m,i} := h(w_m; \xi_{m,i})$, the objective of the VFL framework can be seen as minimizing the following function:

$$F(\boldsymbol{w}; \mathcal{D}, \boldsymbol{Y}) := \frac{1}{D} \sum_{i \in [D]} \mathcal{L}(w_0, h_{1,i}, h_{2,i}, \ldots, h_{M,i}; y_i) \tag{1}$$

where $\mathcal{L}$ is the loss function for a datum $\xi_i$ and its corresponding label $y_i$. For the simplicity of notation, we define the loss function w.r.t a specific model parameter and datum as

$$f(\boldsymbol{w}; \xi_i) := \mathcal{L}(w_0, h_{1,i}, h_{2,i}, \ldots, h_{M,i}; y_i) \tag{2}$$

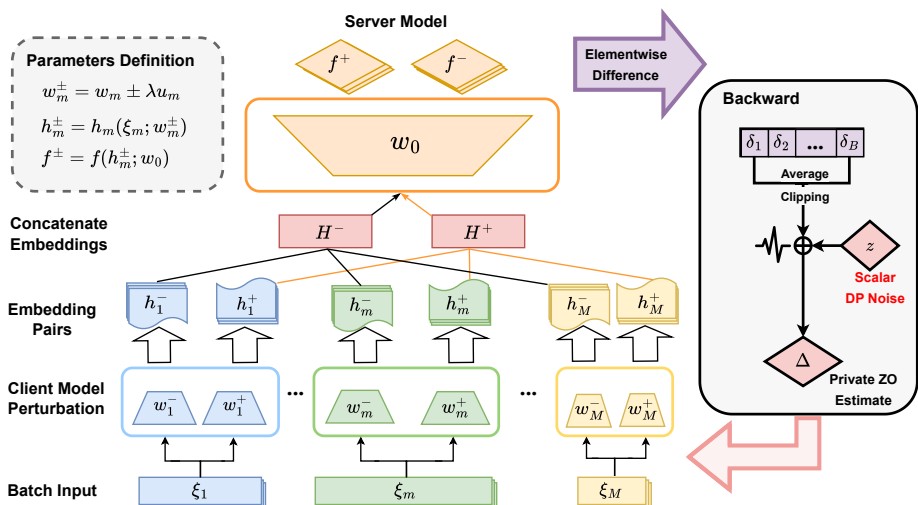

Figure 1: Overview of the training procedure in DPZV. Each client perturbs its local model parameters in two random directions to generate a pair of embeddings, which are then transmitted to the server. The server computes the corresponding function evaluations and applies an elementwise difference to approximate the zeroth-order (ZO) gradient. To ensure differential privacy, scalar-valued Gaussian noise is injected into the aggregated ZO estimate. Unlike traditional vector-valued noise in standard DP algorithms, scalar noise is significantly smaller in norm, thereby preserving model utility even under stringent privacy budgets.

**Memory-Efficient Zeroth-Order Optimizer.** We introduce the two-point gradient estimator (Shamir, 2017), which will serve as our zeroth-order gradient estimator throughout this paper. Let $u$ be uniformly sampled from the Euclidean sphere $\sqrt{d}\mathbb{S}^{d-1}$ and $\lambda > 0$ be the smoothing factor. For a single datum $\xi$ sampled from the whole dataset $\mathcal{D}$, the two-point gradient estimator is defined as

$$g(x;\xi) = \frac{f(x+\lambda u;\xi)-f(x-\lambda u;\xi)}{2\lambda}u \tag{3}$$

To further reduce the memory overhead on client, we adopt the MeZO methodology (Malladi et al., 2023) for our ZO Optimization, which requires only the same memory as the model itself for training. We investigate the memory reduction of this estimator in Appendix G.

**Differential Privacy and DP-SGD.** DP provides a principled framework for protecting individual data in statistical analysis and machine learning. Formally, we adopt the standard $(\epsilon, \delta)$-DP definition:

**Definition 2.1** (($\epsilon, \delta$)-Differential Privacy). A randomized algorithm $\mathcal{M} : \mathcal{X}^n \rightarrow \Theta$ is said to satisfy $(\epsilon, \delta)$-differential privacy if for any neighboring datasets $X, X' \in \mathcal{X}^n$ that differ by only one individual's data, and for any subset of outputs $E \subseteq \Theta$, we have

$$\mathbb{P}[\mathcal{M}(X) \in E] \leq e^{\epsilon}\mathbb{P}[\mathcal{M}(X') \in E] + \delta. \tag{4}$$

One of the most widely adopted private training algorithms is Differentially Private Stochastic Gradient Descent (DP-SGD), which operates by clipping an intermediate result and injecting Gaussian noise during each model update. Specifically, given a minibatch $B_t$ at iteration $t$, DP-SGD performs the following steps: 1) Clip individual gradients: $\bar{g}_i = \nabla\ell(\theta_t, x_i)$, $\tilde{g}_i = \bar{g}_i / \max\left(1, \frac{\|\bar{g}_i\|_2}{C}\right)$. 2) Add Gaussian noise: $\tilde{g}_t = \frac{1}{|B_t|}\left(\sum_{i \in B_t} \tilde{g}_i + \mathcal{N}(0, \sigma^2 C^2 \mathbf{I})\right)$. 3) Update model: $\theta_{t+1} = \theta_t - \eta_t \tilde{g}_t$.

**Challenges and Motivation.** Existing approaches (Chen et al., 2020a; Xie et al., 2024) integrate DP with FO-VFL by injecting DP noise in the forward embedding. In centralized or horizontally federated learning, DP-SGD achieves privacy by adding noise to the backward gradients. However, extending these strategies to ZO optimization introduces new challenges. First, *due to the distributed nature of VFL, DP guarantees must be enforced before the server communicates with clients, so as to prevent malicious clients from mounting label inference attacks.* In both cases, the noise

must match the dimensionality of the protected vector (embedding or gradient), often resulting in high-dimensional noise. This situation is further exacerbated by the second factor: in ZO methods, gradients are approximated through multiple forward evaluations, making them inherently noisy. Adding a *high-dimensional* noise vector as in standard DP-SGD further amplifies this inaccuracy.

A key advantage of adopting ZO optimization, however, lies in its communication efficiency in the backward pass: only a scalar value, rather than a full gradient vector, needs to be exchanged. This observation enables *a more efficient privacy mechanism—injecting scalar noise rather than high-dimensional noise*. Building on this insight, we propose Differentially Private Zeroth-Order Vertical Federated Learning (DPZV), a framework that achieves tunable differential privacy through calibrated scalar noise injection. This design allows for a favorable privacy-utility trade-off in ZO-based VFL, particularly under tight privacy budgets. Unlike first-order VFL methods, which typically add noise to forward embeddings, DPZV introduces noise in the backward pass, targeting the most exploited leakage pathway in VFL: label inference from gradients. A broader discussion of privacy implications is provided in Section 5 and Appendix F.

## 3  METHODOLOGY

In this section, we describe the overall training procedure of DPZV. Based on equation 1, the objective is to collaboratively minimize the global objective function across all clients. The clients hold disjoint features of the same data records, and the server holds the label data. The training procedure proceeds in two iterative steps.

### 3.1  CLIENT UPDATE AND FORWARD COMMUNICATION

The sampled clients compute local information and transmit it to the server. We define the client-server communication as *forward communication*.

**Training Procedure.** Each client $m$ maintaining its own local communication round $t_m$, while the server tracks a global round $t$. Whenever the server receives information from a client, it increments $t$. Upon receiving an update from the server, the client synchronizes by setting $t_m = t$, capturing the latest state. This asynchronous mechanism allows clients to progress without waiting for stragglers, improving throughput and minimizing idle time. The server maintains a copy of the latest local embeddings $\tilde{h}_{m,i}^t$ for all clients $m \in [M]$ and data samples $\xi_i \in \mathcal{D}$. Due to the asynchronous nature of the algorithm, these copies may be stale, as they do not always reflect the most up-to-date model parameters of the clients. Let $\tilde{t}_{m,i}$ denote the client time when the server last updated $\tilde{h}_{m,i}^t$. The delay at server communication round $t$ can then be expressed as $\tau_m^t = t_m - \tilde{t}_{m,i}$, where the delayed model parameters are defined as:

$$\tilde{\chi}^t = \{w_1^{t_1-\tau_1^t}, \ldots, w_M^{t_M-\tau_M^t}\}, \quad \tilde{\boldsymbol{w}}^t = \{w_0^t, \tilde{\chi}^t\}. \tag{5}$$

**Local Embedding Finite Differences.** For each global iteration $t$, a client $m$ is activated, and it samples a mini-batch $\mathcal{B}_m^{t_m} \in \mathcal{D}$ and the corresponding IDs $\mathcal{I}_m^{t_m}$. To approximate gradients via zeroth-order finite differences, client $m$ computes two perturbed local embeddings for the mini-batch,

$$\{h_{m,i}^{t_m+}\}_{i \in \mathcal{I}_m^{t_m}} = \{h(w_m^{t_m} + \lambda_m \boldsymbol{u}_m^{t_m}; \xi_{m,i})\}_{i \in \mathcal{I}_m^{t_m}},$$
$$\{h_{m,i}^{t_m-}\}_{i \in \mathcal{I}_m^{t_m}} = \{h(w_m^{t_m} - \lambda_m \boldsymbol{u}_m^{t_m}; \xi_{m,i})\}_{i \in \mathcal{I}_m^{t_m}} \tag{6}$$

where $\boldsymbol{u}_m^{t_m}$ is sampled uniformly at random from the Euclidean sphere $\sqrt{d_m}\mathbb{S}^{d_m-1}$, and $\lambda_m$ is a smoothing parameter that controls the step size of perturbation. These two perturbed embeddings serve as "positive" and "negative" perturbations of its local parameters, which are forwarded to the server for further computation.

### 3.2  SERVER UPDATE AND BACKWARD COMMUNICATION

The server updates the global model and transmit global updates to the client. We define the server-client communication as *backward communication*.

**Server Side ZO Computation.** After the server receives the local embeddings $h_{m,i}^{t_m}$ from client $m$, it updates its embeddings copy and do computation. For each embedding pair $\{h_{m,i}^{t_m+}, h_{m,i}^{t_m-}\}$, it computes the difference in loss function $\mathcal{L}$ caused by perturbation, divided by the smooth parameter

---

**Algorithm 1:** `DPZV`: Differentially Private Zeroth-Order Vertical Federated Learning

---

**Input** : Data $\mathcal{D}$, batch size $B$, learning rate $\eta_m$, total iteration $T$, smoothing parameter $\lambda_m > 0$, clipping threshold $C > 0$, privacy parameter $\sigma_{dp}$

**Output:** Parameter $w_0, w_m$ for all parties $m \in [M]$

1 Initialize $w_0, w_m$ and set $t, t_m = 0$ for all parties

2 **for** $t = 1, \ldots, T$ **do**

3   Sample ready-to-update client $m \in \{1, \ldots, M\}$.

4   Client $m$ samples a mini-batch $\mathcal{B}_m^{t_m}$ with corresponding IDs $\mathcal{I}_m^{t_m}$.

5   Client $m$ computes perturbed local embeddings $\{h_{m,i}^{t_m+}, h_{m,i}^{t_m-}\}_{i \in \mathcal{I}_m^{t_m}}$ according to equation 6.

6   **(Forward)** Client transmits embeddings to server.

7   Server computes $\Delta_m^t$ according to equation 7 and equation 8, and updates via ZO Optimization.

8   **(Backward)** Client receives $\Delta_m^t$.

9   Client performs local update using equation 9.

10 Client update local time stamp $t_m = t$.

---

$\lambda_m$. Specifically, we define[1]:

$$\delta_{m,i}^{t,t_m} = \frac{\tilde{f}(w_0, h_{m,i}^{t_m+}; y_i) - \tilde{f}(w_0, h_{m,i}^{t_m-}; y_i)}{\lambda_m}, \tag{7}$$

**Scalar DP Noise Injection.** To ensure controllable privacy, the server then clips each $\delta_{m,i}^t$ by a threshold $C$ to bound sensitivity: $\text{clip}_C(\delta_{m,i}^{t,t_m}) = \min\{\delta_{m,i}^{t,t_m}, C\}$. It then samples noise $z_m^{t_m}$ from a Gaussian distribution $\mathcal{N}(0, \sigma_{dp}^2)$. This noise is added to the mean of the per-sample-clipped updates, yielding a differentially private gradient-like quantity:

$$\Delta_m^t = \frac{1}{B} \sum_{i \in \mathcal{I}_m^{t_m}} \text{clip}_C(\delta_{m,i}^{t,t_m}) + z_m^t. \tag{8}$$

The server then performs a backward communication and sends $\Delta_m^t$ to client $m$. The server then updates its global model through two possible operations: $i$) ZO Optimization, $\boldsymbol{w}_0 \leftarrow \boldsymbol{w}_0 - \eta_0 g(w_0; \xi_i)$ (defined in equation 3); or $ii$) stochastic gradient descent (SGD), $\boldsymbol{w}_0 \leftarrow \boldsymbol{w}_0 - \eta_0 \nabla_{w_0} F(\boldsymbol{w}; \xi_i)$,

depending on the constraints on computation resources. The adopted ZO update methodology is explained in section 2.

On the client side, upon receiving $\Delta_m^t$, client $m$ updates its local parameter $\boldsymbol{w}_m$ with learning rate $\eta_m$:

$$\boldsymbol{w}_m \leftarrow \boldsymbol{w}_m - \eta_m \Delta_m^t \boldsymbol{u}_m^{t_m}, \tag{9}$$

where $\boldsymbol{u}_m^{t_m}$ is the same vector used for local perturbation. Our method is well-suited for resource-constrained environments, such as edge devices with limited VRAM or computation power, while still maintaining robust performance and scalability in VFL settings. We summarize the pipeline above in Algorithm 1.

## 4 CONVERGENCE ANALYSIS

In this section, we provide the convergence analysis for `DPZV`. The detailed proofs can be found in Appendix B. For brevity, we define the following notations: $F^t = F(\boldsymbol{w}^t) := F(\boldsymbol{w}^t; \mathcal{D}, \boldsymbol{Y})$, and $f(\boldsymbol{w}; \xi_i)$ as defined in equation 2. We make the following standard assumptions [2]:

---

[1] We slightly abuse the notation, and define $\tilde{f}(w_0, h_{m,i}^{t_m\pm}; y_i) = \mathcal{L}(w_0, \tilde{h}_{1,i}^t, \ldots, h_{m,i}^{t_m\pm}, \ldots, \tilde{h}_{M,i}^t; y_i)$. where we treat $\tilde{f}$ as a function of the server parameter $w_0$ and the perturbed local embeddings.

[2] We claim that assumption 4.1 and 4.2 are standard in VFL and ZO literature (Wang et al., 2024; Castiglia et al., 2023). We follow (Zhang et al., 2024) to make the $\ell$-Lipschitz assumption in order to bound the probability of clipping. Assumption 4.3 is common when dealing with asynchronous participation (Chen et al., 2020a), and can be satisfied when the activations of clients follow independent Poisson processes.

**Assumption 4.1** (Properties of loss function). The VFL objective function $F$ is bounded from below, the function $f(\boldsymbol{w}; \xi_i)$ is $\ell$-Lipschitz continuous and $L$-Smooth for every $\xi_i \in \mathcal{D}$.

**Assumption 4.2** (System boundedness). The following system dynamics are bounded: 1) *Stochastic Noise:* The variance of the stochastic first order gradient is upper bounded in expectation: $\mathbb{E}\left[\|\nabla_{\boldsymbol{w}} f(\boldsymbol{w}; \xi) - \nabla_{\boldsymbol{w}} F(\boldsymbol{w})\|^2\right] \leq \sigma_s^2$. 2) *Time Delay:* The parameter delay $\tau_m^t$ is upper bounded by a constant $\tau$: $\tau^t \leq \tau, \quad \forall m, t$.

**Assumption 4.3** (Independent Participation). Under an asynchronous update system, the probability of one client participating in one communication round is independent of other clients and satisfies: $\mathbb{P}(\text{client } m \text{ uploading}) = q_m$.

We now present the main theorem that provides convergence guarantee for DPZV:

**Theorem 4.4.** *Under assumption 4.1-4.3. Define $\mathcal{F} = \mathbb{E}[F^0 - F^T]$. If we denote $q_* = \min_m q_m$, $d_* = \max_m d_m$ where $d_m$ represent the dimension of model parameters on device $m$, and let all step sizes satisfy: $\eta_0 = \eta_m = \eta \leq \min\{\frac{1}{\sqrt{T d_*}}, \frac{B}{4L(B+8d_0)+8\gamma_1(2d_m+B)}\}$, let the smoothing parameter $\lambda$ satisfy: $\lambda \leq \frac{1}{Ld\sqrt{T}}$, and let the clipping level $C$ satisfy: $C \geq \max\{0, \frac{1}{2}L\lambda d - \ell\sqrt{8\log(2\sqrt{2\pi})}\}$, then for any given global iteration $T \geq 1$, we have the following upper bound on the gradient norm:*

$$\frac{1}{T}\sum_{t=0}^{T-1} \mathbb{E}\left[\|\nabla_{\boldsymbol{w}} F(\boldsymbol{w}^t)\|^2\right] \leq \mathcal{O}\left(\frac{\mathcal{F}\sqrt{d_*}}{\sqrt{T}} + \frac{d_*}{T} + \frac{(\sigma_s^2/B + \sigma_{dp}^2)\sqrt{d_*}}{\sqrt{T}} + \frac{C^2\sqrt{d_*}}{(\exp(C^2)-1)B\sqrt{T}}\right), \quad (10)$$

**Discussion.** The first term is influenced by the model's initialization, $F^0$. This term also enjoys the same rate as ZO optimization in the centralized case (Ghadimi & Lan, 2013). The second term $\mathcal{O}(\frac{d_*}{T})$ is a standard term for ZOO methods based on the usage of ZO estimator updates. The third term captures the impact of various noise sources in the learning system. Here, $\sigma_s^2/B$ represents the noise introduced by stochastic gradients, and $\sigma_{dp}^2$ corresponds to the variance of the injected DP noise. While reducing $\sigma_{dp}^2$ improves utility, it comes at the cost of weakening privacy guarantees. Consequently, this term encapsulates the fundamental trade-off between model performance, computational cost, and privacy budget. The fourth term, quantifies the impact of the gradient clipping operation on convergence. As $C$ increases, increases, this term diminishes, effectively recovering the non-private case in the limit. However, the noise variance $\sigma_{dp}^2$ also scales linearly with $C$, which can impede overall convergence. This trade-off underscores the importance of carefully tuning the sensitivity level to balance privacy preservation and learning efficiency.

Based on the privacy analysis in section 5, we further substitute the level of DP variance $\sigma_{dp} = \frac{2C\sqrt{T}}{D\mu}$ into this convergence rate. By selecting $T = \mathcal{O}\left(\frac{D^2\mu^2\sqrt{d}}{C^2}\left(\mathcal{F} + \frac{\sigma^2}{B} + \frac{C^2}{e^{C^2}-1}\right)\right)$, the overall convergence upper bound becomes $\mathcal{O}\left(\frac{\sqrt{d}}{D\mu}\right)$, which matches the known rate of DP-SGD for *first-order* methods (Chen et al., 2020b). This result underscores a key insight of our analysis: although zeroth-order optimization typically converges more slowly than its first-order counterpart in non-private settings, it can achieve the same convergence rate when subject to differential privacy constraints.

## 5 PRIVACY ANALYSIS

While data are kept local in the VFL framework, the communication of backward gradient during the training process, as well as the exposure of model weights, can pose threat to sensitive information (Papernot et al., 2018). In this section, we talk about how our algorithm protects privacy under these threats. We consider the *honest-but-curious* threat model, where participants adhere strictly to the protocols of VFL without deviating from agreed procedures. However, they may attempt to infer private information from intermediate results exchanged during training.

**Theorem 5.1.** *Under assumption 4.1-4.3, suppose the privacy parameter $\sigma_{dp}$ is $\sigma_{dp} = \frac{2C\sqrt{T}}{D\mu}$, where $D$ denotes the volume of the dataset, $T$ defines total iterations, and $\mu > 0$ controls the privacy level. The training process of Algorithm 1 is seen to be $(\epsilon, \delta(\epsilon))$-differential private for $\forall \epsilon > 0$, where*

$$\delta(\epsilon) = \Phi(-\frac{\epsilon}{\mu} + \frac{\mu}{2}) - e^\epsilon\Phi(-\frac{\epsilon}{\mu} - \frac{\mu}{2}) \quad (11)$$

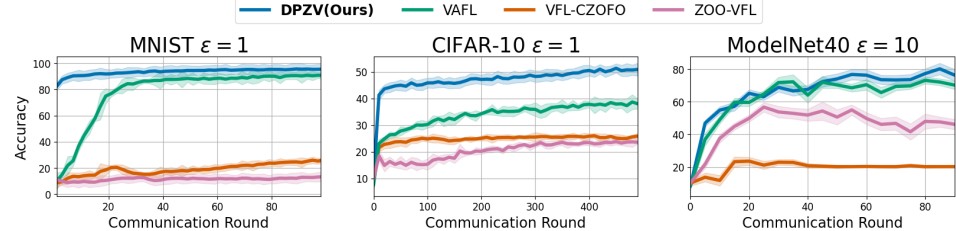

(a) Test Accuracy of VFL Methods on image classification tasks under DP constraints. $\delta$ is set to $1 \times 10^{-3}$. DPZV outperforms first-order VFL methods on two datasets and surpasses all other ZO-based methods across all three datasets, showing both a higher accuracy and a faster convergence rate.

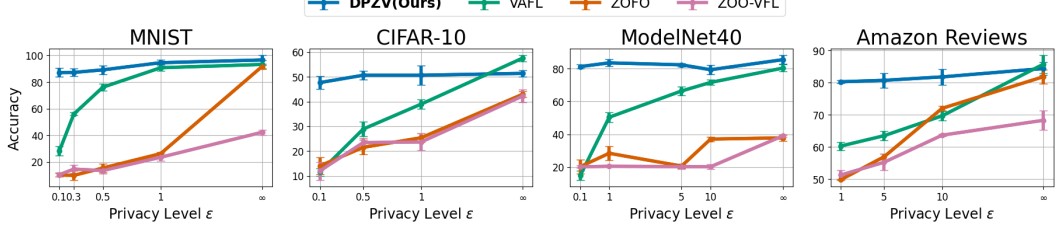

(b) Privacy-Accuracy tradeoff across different datasets and algorithms. We use a constant level of $\delta = 1 \times 10^{-3}$ and vary $\epsilon$ to simulate different privacy levels. Our algorithm consistently outperforms baselines under tight privacy budget, showing a slower decay in performance than baselines as $\epsilon$ decreases.

**Discussion.** The theorem provides privacy guarantee under the "honest-but-curious" threat model, where one or a few malicious clients try to do inference attacks by collecting information of the system. By providing only the differentially private ZO information $\Delta_m^t$, the algorithm protects for labels (Fu et al., 2022) and per-record influence, because the attacker cannot differentiate a single datum in the dataset $\mathcal{D}$ and label set $Y$. The differential privacy parameter $\mu$ is related to $(\epsilon, \delta)$ through the relationship defined in equation 11. Given any two of these parameters, the third can be determined by solving equation 11, allowing full flexibility in controlling the privacy level.

Although the forward embeddings $h_{m,i}$ are not protected by differential privacy, we note that label information is only exposed in the backward pass. The following corollary formalizes

**Corollary 5.2.** *Under the same condition of Theorem 5.1, at any time $t$ of the training process and any client $m$, the forward embedding $h_{m,i}^t$ is differentially private w.r.t. the labels.*

A detailed discussion of our framework's robustness against more threat models is provided in appendix F, and the detailed proofs can be found in appendix C.

## 6 EXPERIMENTS

**Dataset and Baselines.** We consider four datasets: image dataset MNIST (Deng, 2012), CIFAR-10 (Krizhevsky et al., 2009), semantic dataset Amazon Review Polarity (McAuley & Leskovec, 2013), and multi-view dataset ModelNet40 (Wu et al., 2015). For each dataset, we conduct a grid search on learning rates and other hyperparameters. We run 100 epochs on each method and select the best validation model. We run each algorithm under three random seeds and compute the sample variance for the generosity of our result. Additional information on the selection of dataset, data processing and model architecture can be found in Appendix E.

We compare our algorithm against several SotA VFL methods: 1) VAFL (Chen et al., 2020a) 2) ZOO-VFL (Zhang et al., 2021), 3) VFL-CZOFO (Wang et al., 2024). All methods assume that the server holds the labels, and concatenates the embeddings of clients as the input of the server. VAFL updates its model through first-order optimization in an asynchronous manner, and achieves DP by adding random noise to the output of each local embedding. We use VAFL as a first-order baseline of VFL method. Contrary to VAFL, ZOO-VFL and ZOFO both adopt ZO optimization in their training procedure. ZOO-VFL substitutes the first order optimization by ZO optimization in common VFL methods. VFL-CZOFO uses a cascade hybrid optimization method that computes the intermediate gradient via ZOO, while keeping the back propagation on both server and client. To enforce DP and compare all methods fairly, we follow the approach in VAFL (Chen et al., 2020a), applying the same DP mechanism to both VFL-CZOFO and ZOO-VFL, which involves clipping the embeddings and adding calibrated vector noise. In this paper, we only focus on the same update behavior as VAFL which updates server and client once for every communication round. In addition, model delay has

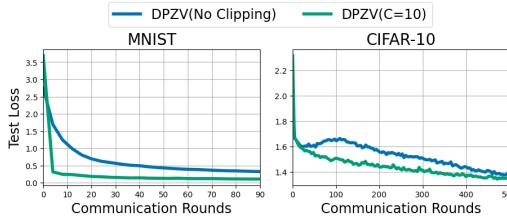
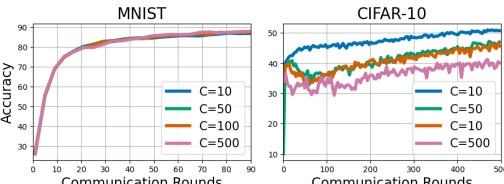

Figure 3: Effect of gradient clipping on `DPZV` under non-private settings. We compare models trained with and without clipping of the ZO information. Across both MNIST and CIFAR-10, clipping accelerates convergence and stabilizes training.

Figure 4: Effect of clipping threshold on `DPZV` under differential privacy constraint $\epsilon = 1$. While all clipping levels perform similarly on MNIST, smaller thresholds ($C = 10$) significantly improve accuracy and convergence on CIFAR-10.

been manually adjusted for all methods based on the per-batch computation time on a single client to simulate client heterogeneity and ensure a fair comparison.

**Accelerated convergence of DPZV with privacy guarantees.** Figure 2a presents the performance evaluation of `DPZV` against all baselines on image classification tasks under certain DP levels. The results show that `DPZV` consistently outperforms all baselines on MNIST and CIFAR-10 under strict privacy budget ($\epsilon = 1, \delta = 1 \times 10^{-3}$). On ModelNet40, where training is more challenging due to larger models and a greater number of clients, we slightly relax the DP constraint ($\epsilon = 10$), and observe that `DPZV` maintains a competitive advantage over other ZO-based methods while achieving accuracy comparable to the first-order baseline `VAFL`. The scalar noise injected in `DPZV` effectively constrains the noise magnitude, mitigating the instability typically introduced by noisy zeroth-order gradient estimators. This leads to more stable training dynamics than other ZO based methods. Moreover, in contrast to the high-dimensional vector noise used in first-order baseline `VAFL`, scalar noise incurs significantly less loss, resulting in superior performance under stringent privacy budgets.

**DPZV elevates privacy-utility tradeoff.** Figure 2b presents the accuracy-privacy tradeoff across four benchmark datasets: MNIST, CIFAR-10, ModelNet40, and Amazon Reviews. We evaluate the robustness of our proposed method `DPZV` under various privacy budgets $\epsilon$, while fixing the failure probability $\delta = 1 \times 10^{-3}$. Across all tasks, `DPZV` consistently achieves the highest accuracy under tight privacy regimes ($\epsilon \leq 1$), indicating its ability to maintain model utility even with stringent differential privacy constraints. Notably, on MNIST and ModelNet40, `DPZV` shows only a marginal drop in accuracy as $\epsilon$ decreases, while the competing methods suffer substantial degradation. For instance, on CIFAR-10 at $\epsilon = 0.1$, `DPZV` maintains over 90% accuracy, whereas `ZOFO` and `ZOO-VFL` fall below 30%. Similar trends are observed on Amazon Reviews, where `DPZV` consistently outperforms baselines under strict privacy, especially at $\epsilon = 1$ and $\epsilon = 5$. These results validate the effectiveness of our scalar-noise-based DP mechanism in balancing utility and privacy, and *highlight its advantage in real-world privacy-sensitive federated learning scenarios*.

**Clipping benefits convergence.** We observe in Figure 2b that even under no privacy constraints ($\epsilon = \infty$), `DPZV` outperforms other ZO based algorithms. A key distinction lies in `DPZV`'s use of scalar clipping on ZO information, while originally introduced for differential privacy, also acts as a form of gradient regularization. This regularization effect has been shown to improve convergence in prior work (Zhang et al., 2019), and we observe similar benefits here. Figure 3 verifies our insight by comparing the convergence behavior of `DPZV` with and without gradient clipping under a non-private setting. The plots show test loss versus communication rounds. In both datasets, applying clipping to the ZO information significantly improves convergence speed. For MNIST, clipped `DPZV` reaches low test loss much faster and stabilizes more smoothly. On CIFAR-10, the clipped version also demonstrates consistently lower loss throughout training. These results suggest that clipping not only stabilizes training but also enhances convergence efficiency, even when privacy is not enforced.

**Impact of Sensitivity Level.** Figure 4 presents the performance of `DPZV` under different sensitivity levels, which is controlled by the clipping threshold $C$. We apply a fixed privacy budget of $\epsilon = 1, \delta = 1 \times 10^{-3}$ on both MNIST and CIFAR-10. In the MNIST setting, all clipping levels achieve similar convergence and final accuracy, suggesting the model is robust to the choice of $C$ in simple tasks. In contrast, the CIFAR-10 results highlight a pronounced impact: smaller clipping values

(e.g., $C = 10$) yield better accuracy and stability over training. Larger thresholds, such as $C = 500$ degrade performance, likely due to the excessive noise required to satisfy DP constraints. These results emphasize the importance of careful clipping calibration in more complex settings to ensure a good privacy-utility tradeoff.

# 7 RELATED WORK

We now discuss related work along two dimensions: (i) vertical federated learning and its privacy-preserving variants, and (ii) zeroth-order optimization.

**Vertical Federated Learning.** Vertical Federated Learning (VFL) enables collaborative training across organizations with vertically partitioned features. Early VFL frameworks focused on simple client-side models such as logistic regression and linear models (Hardy et al., 2017). These methods prioritized simplicity, but lacked expressiveness for complex tasks. To address this limitation, larger client-side models like deep neural networks (DNNs) were adopted (Chen et al., 2020a; Castiglia et al., 2023; Xie et al., 2024).

A key challenge in VFL is the communication overhead incurred during training. One popular mechanism for reducing communication overhead is by allowing for multiple local updates in-between aggregations. In this respect, (Liu et al., 2022) introduced FedBCD, which allows clients to perform multiple gradient iterations before synchronization. Similarly, `Flex-VFL` (Castiglia et al., 2023) proposed a flexible strategy offering varying local update counts per party, constrained by a communication timeout. `VIMADMM` (Xie et al., 2024) adopted an ADMM-based approach to enable multiple local updates in VFL. On the other hand, Asynchronous VFL methods (e.g., `FDML` (Hu et al., 2019), `VAFL` (Chen et al., 2020a)) decouple coordination, allowing clients to update models independently, thus improving scalability. However, in FO methods, the backward pass through neural networks typically imposes communication overhead, while our ZO-based approach significantly reduces the cost associated with backward propagation.

Privacy guarantee is another critical challenge for VFL adoption. Some VFL architectures use crypto-based privacy-preserving techniques such as Homomorphic Encryption (HE) (Cheng et al., 2021), but lack formal assurances. In contrast, DP provides rigorous mathematical protection. Key DP-based methods include `VAFL` (Chen et al., 2020a), which injects Gaussian noise into client embeddings during forward propagation to achieve Gaussian DP, and `VIMADMM` (Xie et al., 2024), which perturbs linear model parameters with bounded sensitivity, ensuring DP guarantees for convex settings. Our work overcomes challenges in developing such methods for the ZO setting.

**Zeroth-Order Optimization.** Recent research has explored Zeroth-Order (ZO) optimization within VFL to accommodate resource-constrained clients with non-differentiable models and to reduce gradient leakage. Early work like `ZOO-VFL` (Zhang et al., 2021) adopted a naive ZO approach throughout VFL training but provided no DP guarantees. `VFL-CZOFO` (Wang et al., 2024) introduced a cascaded hybrid optimization method, combining zeroth-order and first-order updates, which leveraged intrinsic noise from ZO for limited privacy. However, its DP level was not adjustable, resulting in insufficient protection.

More recently, `MeZO` (Malladi et al., 2023) proposed a memory-efficient ZO algorithm. Building upon these ideas, `DPZero` (Zhang et al., 2024) and `DPZO` (Tang et al., 2024) introduced private ZO variants offering baseline privacy features. However, these methods are designed for centralized settings and cannot be directly extended to the VFL paradigm. Extending beyond previous efforts to combine ZO optimization with VFL, we further integrate a controllable differential privacy mechanism, achieving an elevated trade-off between privacy and model performance.

# 8 CONCLUSION

In this work, we propose `DPZV`, the first zero-order VFL framework that achieves robust and controllable privacy guarantees against inference attacks while maintaining strong model utility. Through rigorous theoretical analysis, we establish the strong convergence properties of our algorithm. Extensive experiments demonstrate that `DPZV` outperforms all baselines across various privacy levels and cost evaluations. These results provide valuable insights for future advancements in VFL and distributed ZO optimization methods.

**Reproducibility Statement** All resources required to replicate our main experimental findings are provided. The datasets employed are publicly accessible, while details on the model architecture, training procedure, and hyperparameter choices can be found in Sec. 6 and Appendix E. In addition, the implementation code is supplied in the supplementary material.

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

# Appendix

## A PRELIMINARIES AND OUTLINE

We first define the following notation table to facilitate the proof:

| Notation | Description |
|---|---|
| $\boldsymbol{w} = [w_0, w_1, w_2, \ldots, w_M]$ | All learnable parameters |
| $d_0, d_1, \ldots, d_M$ | Dimension of parameters on server (machine 0) and client $1, \ldots, M$ |
| $d = \sum_{m=0}^{M} d_m$ | Dimension of all parameters |
| $f(\boldsymbol{w}; \xi_i) := \mathcal{L}(w_0, h_{1,i}, h_{2,i}, \ldots, h_{M,i}; y_i)$ | Loss function with regard to datum with ID $i$ |
| $F(\boldsymbol{w}) := F(\boldsymbol{w}; \mathcal{D}, \boldsymbol{Y})$ | Global loss function |
| $\chi^t = [w_1^{t_1}, \ldots, w_M^{t_M}]$ | Latest learnable parameters of all clients at server time $t$ |
| $\tilde{\chi}^t = [w_1^{t_1 - \tau_1^t}, \ldots, w_M^{t_M - \tau_M^t}]$ | Delayed learnable parameters of all clients at server time $t$ |
| $\boldsymbol{w}^t = [w_0^t, w_1^{t_1}, \ldots, w_M^{t_M}]$ | Latest learnable parameters of all clients and the server at server time $t$ |
| $\tilde{\boldsymbol{w}}^t = [w_0^{t - \tau_1^t}, w_1^{t_1 - \tau_1^t}, \ldots, w_M^{t_M - \tau_M^t}]$ | Delayed learnable parameters of all clients and the server at server time $t$ |
| $h_{m,i}^{t_m \pm} = h_m(w_m^{t_m} \pm \lambda_m \boldsymbol{u}_m^{t_m}; \xi_{m,i})$ | Local embeddings of client $m$ for data sample $i$ at client time $t_m$ under the perturbed parameters |
| $\delta_{m,i}^{t, t_m}$ as defined in equation 7 | zeroth-order difference information from client $m$ and data sample $i$ at server time $t$ and client time $t_m$ |
| $g_{m,i}^t(\tilde{\boldsymbol{w}}^t) = \delta_{m,i}^{t, t_m} \boldsymbol{u}_m^{t_m}$ | zeroth-order gradient estimator from client $m$ and data sample $i$ at server time $t$ (with delay) |
| $\breve{g}_{m,i}^t(\tilde{\boldsymbol{w}}^t) = \mathrm{clip}_C\left(\delta_{m,i}^{t, t_m}\right) \boldsymbol{u}_m^{t_m}$ | Clipped zeroth-order gradient estimator from client $m$ and data sample $i$ at server time $t$ (with delay) |
| $\breve{G}_m^t(\tilde{\boldsymbol{w}}^t) = (1/B) \sum_{i \in \mathcal{I}_m^{t_m}} \breve{g}_{m,i}^t(\tilde{\boldsymbol{w}}^t) + z_m^{t_m} \boldsymbol{u}_m^{t_m}$ | Clipped differential private zeroth-order gradient estimator from client $m$ at server time $t$ (with delay) |
| $G_m^t(\tilde{\boldsymbol{w}}^t) = (1/B) \sum_{i \in \mathcal{I}_m^{t_m}} g_{m,i}^t(\tilde{\boldsymbol{w}}^t) + z_m^{t_m} \boldsymbol{u}_m^{t_m}$ | Non-clipped differential private zeroth-order gradient estimator from client $m$ at server time $t$ (with delay) |

Table 1: Table of Notations

Note that in the notation table, we use "˘" to define clipped gradient estimators, and we use "˜" to denote delayed model parameters. In the rest of the proof, we also use gradient estimators parameterized by the no delaying parameters $\boldsymbol{w}$ instead of $\tilde{\boldsymbol{w}}$ to assume that we update the model without delay. To begin with, we restate the assumptions required for establishing the convergence analysis.

**Assumption A.1** ($\ell$-Lipschitz). The function $f(\boldsymbol{w}; \xi)$ is $\ell$-Lipschitz continuous for every $\xi$.

**Assumption A.2** ($L$-Smooth). The function $f(\boldsymbol{w}; \xi)$ is $L$-Smooth for every $\xi$. Specifically, there exists an $L > 0$ for all $m = 0, \ldots, M$ such that $\|\nabla_{w_m} f(\boldsymbol{w}) - \nabla_{w_m} f(\boldsymbol{w}')\| \le L \|\boldsymbol{w} - \boldsymbol{w}'\|$.

**Assumption A.3** (Bounded gradient variance). The variance of the stochastic first order gradient is upper bounded in expectation:

$$\mathbb{E}\left[\|\nabla_{\boldsymbol{w}} f(\boldsymbol{w}; \xi) - \nabla_{\boldsymbol{w}} F(\boldsymbol{w})\|^2\right] \le \sigma_s^2$$

**Assumption A.4** (Independent Participation). The probability of one client participating in one communication round is independent of other clients and satisfies

$$\mathbb{P}(\text{client } m \text{ uploading}) = q_m$$

Specially, we set $q_0 = 1$ as the server always participates in the update.

One of the important parts in the proof of Theorem 4.4 is to bound the zeroth-order gradient estimator. We first introduce the formal definition of zeroth-order two-point gradient estimator that is used in our algorithm, and prove some technical lemmas that reveal some important properties.

**Definition A.5.** Let $\boldsymbol{u}$ be uniformly sampled from the Euclidean sphere $\sqrt{d}\mathbb{S}^{d-1}$. For any function $f(x): \mathbb{R}^d \to \mathbb{R}$ and $\lambda > 0$, we define its zeroth-order gradient estimator as

$$g(\boldsymbol{w}) = \frac{(f(\boldsymbol{w} + \lambda\boldsymbol{u}) - f(\boldsymbol{w} - \lambda\boldsymbol{u}))}{2\lambda}\boldsymbol{u} \tag{12}$$

**Lemma A.6.** Let $g(\boldsymbol{w})$ be the zeroth-order gradient estimator defined as in equation 12, with $f(\boldsymbol{w})$ being the loss function. We define the smoothed function $f_\lambda(\boldsymbol{w}) = \mathbb{E}_{\boldsymbol{u}}[f(\boldsymbol{w} + \lambda\boldsymbol{u})]$, where $v$ is uniformly sampled from the Euclidean ball $\sqrt{d}\mathbb{B}^d = \{\boldsymbol{w} \in \mathbb{R}^d \mid \|\boldsymbol{w}\| \leq \sqrt{d}\}$. The following properties hold:

(i) $f_\lambda(\boldsymbol{w})$ is differentiable and $\mathbb{E}_{\boldsymbol{u}}[g(\boldsymbol{w})] = \nabla f_\lambda(\boldsymbol{w})$.

(ii) If $f(\boldsymbol{w})$ is L-smooth, we have that

$$\|\nabla f(\boldsymbol{w}) - \nabla f_\lambda(\boldsymbol{w})\| \leq \frac{L}{2}\lambda d^{3/2}, \tag{13}$$

$$|f(\boldsymbol{w}) - f_\lambda(\boldsymbol{w})| \leq \frac{L}{2}\lambda^2 d, \tag{14}$$

and

$$\mathbb{E}_{\boldsymbol{u}}[\|g_\lambda(\boldsymbol{w})\|^2] \leq 2d \cdot \|\nabla f(\boldsymbol{w})\|^2 + \frac{L^2}{2}\lambda^2 d^3. \tag{15}$$

Based on equation 14, we can further show:

$$\|\nabla f_\lambda(\boldsymbol{w})\|^2 \leq 2\|\nabla f(\boldsymbol{w})\|^2 + \frac{L^2}{2}\lambda^2 d^3 \tag{16}$$

$$\|\nabla f(\boldsymbol{w})\|^2 \leq 2\|\nabla f_\lambda(\boldsymbol{w})\|^2 + \frac{L^2}{2}\lambda^2 d^3 \tag{17}$$

This is the standard result of zeroth-order optimization. The proof of the Lemma is given by (Nesterov & Spokoiny, 2017) We also find the following lemmas useful in the proof:

**Lemma A.7.** Let $\boldsymbol{u}$ be uniformly sampled from the Euclidean sphere $\sqrt{d}S^{d-1}$, and $\boldsymbol{a}$ be any vector of constant value. We have that $\mathbb{E}[\boldsymbol{u}] = 0$ and

$$\mathbb{P}(|\boldsymbol{u}^\top\boldsymbol{a}| \geq C) \leq 2\sqrt{2\pi}\exp\left(-\frac{C^2}{8\|\boldsymbol{a}\|^2}\right).$$

*Proof.* This lemma follows exactly from Lemma C.1. in (Zhang et al., 2024). □

**Lemma A.8.** Let $Q$ be the event that clipping happened for a sample $\xi$, $d$ be the model dimension, and $L, \ell$ be the Lipschitz and smooth constant as defined in assumption A.1 and assumption A.2. For $\forall C_0 > 0$, if the clipping threshold $C$ follows $C \geq C_0 + L\lambda d/2$, we have the following upper bound for the probability of clipping:

$$P = \mathbb{P}(Q) \leq 2\sqrt{2\pi}\exp(-\frac{C_0^2}{8\ell^2}) = \Xi \tag{18}$$

*Proof.* Since $f(\boldsymbol{u}; \xi)$ is L-Smooth for every $\xi$, we have

$$\frac{|f(\boldsymbol{w} + \lambda\boldsymbol{u}; \xi) - f(\boldsymbol{w} - \lambda\boldsymbol{u}; \xi)|}{2\lambda} \leq |u^\top\nabla f(\boldsymbol{u}; \xi)| + \frac{|f(\boldsymbol{w} + \lambda\boldsymbol{u}; \xi) - f(\boldsymbol{w}; \xi) - \lambda\boldsymbol{u}^\top\nabla f(\boldsymbol{w}; \xi)|}{2\lambda}$$

$$+ \frac{|f(\boldsymbol{w} - \lambda\boldsymbol{u}; \xi) - f(\boldsymbol{w}; \xi) + \lambda\boldsymbol{u}^\top\nabla f(\boldsymbol{w}; \xi)|}{2\lambda}$$

$$\leq |\boldsymbol{u}^\top \nabla f(\boldsymbol{w};\xi)| + \frac{L}{2}\lambda.$$

Therefore, by Lemma A.7 and Assumption A.1, we obtain

$$
\begin{aligned}
\mathbb{P}(Q) &= \mathbb{P}\left(\frac{|f(\boldsymbol{w}+\lambda\boldsymbol{u};\xi_i) - f(\boldsymbol{w}-\lambda\boldsymbol{u};\xi_i)|}{2\lambda} \geq C_0 + \frac{L}{2}\lambda\right) \\
&\leq \mathbb{P}(|\boldsymbol{u}^\top \nabla f(\boldsymbol{w};\xi_i)| \geq C_0) \\
&\leq 2\sqrt{2\pi}\exp\left(-\frac{C_0^2}{8\|\nabla f(\boldsymbol{w};\xi_i)\|^2}\right) \\
&\leq 2\sqrt{2\pi}\exp\left(-\frac{C_0^2}{8\ell^2}\right).
\end{aligned}
$$

$\square$

**Lemma A.9** (Expectation and Variance of Clipped Zeroth-order Gradient Estimator). *Recall that $\breve{g}_{m,i}^t(\boldsymbol{w}^t)$ is defined as the clipped zeroth-order gradient estimator assuming no communication delay, random perturbation $\boldsymbol{u}$ is defined in Lemma A.7, and event $Q$ is defined in Lemma A.8. We have the following properties:*

$(i)$ *When taking expectation w.r.t $\boldsymbol{u}$ and $Q$, the clipped zeroth-order gradient estimator follows*

$$\mathbb{E}_{\boldsymbol{u}}[\breve{g}_{m,i}^t(\boldsymbol{w}^t)] = (1-P)\nabla_{w_m}F_\lambda(\boldsymbol{w}^t) \tag{19}$$

$(ii)$ *The variance of $\breve{g}_{m,i}^t(\boldsymbol{w}^t)$ follows*

$$\mathrm{Var}(\breve{g}_{m,i}^t(\boldsymbol{w}^t)) \leq (1-P)(2d_m\|\nabla_{w_m}F(\boldsymbol{w}^t)\|^2 + 2d_m\sigma_s^2 + \frac{L^2}{2}\lambda^2 d_m^3) + PC^2 d_m - (1-P)^2\|\nabla_{w_m}F_\lambda(\boldsymbol{w}^t)\|^2 \tag{20}$$

*Proof.* For $(i)$, we have

$$
\begin{aligned}
\mathbb{E}\left[\breve{g}_{m,i}^t(\boldsymbol{w}^t)\right] &= \mathbb{E}\left[\breve{g}_{m,i}^t(\boldsymbol{w}^t)|\bar{Q}\right]\mathbb{P}(\bar{Q}) + \mathbb{E}\left[\breve{g}_{m,i}^t(\boldsymbol{w}^t)|Q\right]\mathbb{P}(Q) \\
&= \mathbb{E}\left[\tilde{g}_{m,i}^t(\boldsymbol{w}^t)\right](1-\mathbb{P}(Q)) + \mathbb{E}\left[C\boldsymbol{u}_m^t\right]\mathbb{P}(Q) \\
&= (1-P)\nabla_{w_m}F_\lambda(\boldsymbol{w}^t)
\end{aligned}
$$

where in the first step we applied the Law of Total Expectation, and in the last step we used the property $(i)$ in Lemma A.6 and $(i)$ in Lemma A.7.

By equation 19, we can further bound the variance of $\breve{g}_{m,i}^t$

$$
\begin{aligned}
&\mathrm{Var}(\breve{g}_{m,i}^t(\boldsymbol{w}^t)) \\
&= \mathbb{E}\left[\left\|\breve{g}_{m,i}^t(\boldsymbol{w}^t) - (1-P)\nabla_{w_m}F_\lambda(\boldsymbol{w}^t)\right\|^2\right] \\
&= \mathbb{E}\left[\left\|\breve{g}_{m,i}^t(\boldsymbol{w}^t)\right\|^2\right] - (1-P)^2\|\nabla_{w_m}F_\lambda(\boldsymbol{w}^t)\|^2 \\
&= \mathbb{E}\left[\left\|\tilde{g}_{m,i}^t(\boldsymbol{w}^t)\right\|^2|\bar{Q}\right]\mathbb{P}(\bar{Q}) + \mathbb{E}\left[C^2\|\boldsymbol{u}_m^t\|^2|Q\right]\mathbb{P}(Q) - (1-P)^2\|\nabla_{w_m}F_\lambda(\boldsymbol{w}^t)\|^2 \\
&\overset{1)}{\leq} (1-P)(2d_m\|\nabla_{w_m}f(\boldsymbol{w}^t;\xi_{m,t})\|^2 + \frac{L^2}{2}\lambda^2 d_m^3) \\
&\quad + PC^2 d_m - (1-P)^2\|\nabla_{w_m}F_\lambda(\boldsymbol{w}^t)\|^2 \\
&\overset{2)}{\leq} (1-P)(2d_m\|\nabla_{w_m}F(\boldsymbol{w}^t)\|^2 + 2d_m\sigma_s^2 + \frac{L^2}{2}\lambda^2 d_m^3) \\
&\quad + PC^2 d_m - (1-P)^2\|\nabla_{w_m}F_\lambda(\boldsymbol{w}^t)\|^2
\end{aligned}
$$

where 1) is by the property of zeroth-order gradient estimator equation 15 and 2) follows from the bounded gradient assumption(Assumption A.3). $\square$

**Lemma A.10** (Bounds on the variance of DP Zeroth-order Gradient Estimator). *Let $\breve{G}_m^t(\boldsymbol{w}^t)$ be the Differential Private Zeroth-order Gradient without delay. Under the same condition as Lemma A.6, we can bound the variance of $\breve{G}_m^t(\boldsymbol{w}^t)$ in expectation, with the expectation taken on random direction $\boldsymbol{u}$, DP noise $z$, and clipping event $Q$:*

$$\operatorname{Var}(\breve{G}_m^t(\boldsymbol{w}^t)) \leq \frac{1-P}{B}(2d_m \mathbb{E}\left[\|\nabla_{w_m} F(\boldsymbol{w}^t)\|^2\right] + 2d_m \sigma_s^2 + \frac{L^2}{2}\lambda^2 d_m^3) + \frac{P}{B}C^2 d_m$$
$$- \frac{(1-P)^2}{B}\mathbb{E}\left[\|\nabla_{w_m} F_\lambda(\boldsymbol{w}^t)\|^2\right] + \sigma_{dp}^2 d_m \tag{21}$$

*Proof.* First we show that the expectation on $\breve{G}_m^t(\boldsymbol{w}^t)$ can be written as:

$$\mathbb{E}_u[\breve{G}_m^t(\boldsymbol{w}^t)] = \frac{1}{B}\sum_{i\in\mathcal{I}_m^{t_m}}\mathbb{E}\left[\breve{g}_{m,i}^t(\boldsymbol{w}^t)\right] + \mathbb{E}\left[z_m^t \boldsymbol{u}_m^t\right] = (1-P)\nabla_{w_m}F_\lambda(\boldsymbol{w}^t)$$

Thus, the variance can be bounded by:

$$\operatorname{Var}(\breve{G}_m^t(\boldsymbol{w}^t))$$
$$= \mathbb{E}\left[\left\|\breve{G}_m^t(\boldsymbol{w}^t) - (1-P)\nabla_{w_m}F_\lambda(\boldsymbol{w}^t)\right\|^2\right]$$
$$= \mathbb{E}\left[\left\|\frac{1}{B}\sum_{i\in\mathcal{I}_m^{t_m}}\left(\breve{g}_{m,i}^t(\boldsymbol{w}^t) - (1-P)\nabla_{w_m}F_\lambda^t(\boldsymbol{w}^t)\right) + z_m^t \boldsymbol{u}_m^t\right\|^2\right]$$
$$= \frac{1}{B^2}\sum_{i\in\mathcal{I}_m^{t_m}}\mathbb{E}\left[\left\|\breve{g}_{m,i}^t(\boldsymbol{w}^t) - (1-P)\nabla_{w_m}F_\lambda^t(\boldsymbol{w}^t)\right\|^2\right] + \mathbb{E}\left[\|z_m^t \boldsymbol{u}_m^t\|^2\right]$$
$$\overset{(a)}{\leq} \frac{1-P}{B}(2d_m\mathbb{E}\left[\|\nabla_{w_m}F(\boldsymbol{w}^t)\|^2\right] + 2d_m\sigma_s^2 + \frac{L^2}{2}\lambda^2 d_m^3) + \frac{P}{B}C^2 d_m - \frac{(1-P)^2}{B}\mathbb{E}\left[\|\nabla_{w_m}F_\lambda(\boldsymbol{w}^t)\|^2\right] + \sigma_{dp}^2 d_m,$$

where $(a)$ follows from equation 20. $\square$

Finally, for analyzing the delayed information in model parameter, we define the following Lyapunov function:

$$V^t = F(\boldsymbol{w}^t) + \sum_{i=1}^{\tau}\gamma_i\|\chi^{t+1-i} - \chi^{t-i}\|^2 \tag{22}$$

with $\gamma_i$ to be determined later.

# B    PROOF OF CONVERGENCE(THEOREM 4.4)

We first state the full version of the main convergence theorem 4.4.

**Theorem B.1.** *Under Assumption A.1 to A.4, and assume the client delay $\tau_m$ is uniformly upper bounded by $\tau$. If we denote $q_* = \min_m q_m$, $d_* = \max_m d_m$, and let $\eta_0 = \eta_m = \eta \leq \min\{\frac{1}{\sqrt{T}d_*}, \frac{B}{4L(B+8d_0)+8\gamma_1(2d_m+B)}\}$, $\lambda \leq \frac{1}{Ld\sqrt{T}}$, we have the following theorem:*

$$\frac{1}{T}\sum_{t=0}^{T-1}\mathbb{E}\left[\|\nabla_{\boldsymbol{w}}F(\boldsymbol{w}^t)\|^2\right] \leq \frac{1}{1-\Xi}\left\{\frac{8\sqrt{d_*}}{q_*T^{1/2}}\mathbb{E}\left[F^0 - F^T\right] + \frac{2d_*}{q_*T} + \frac{8\sqrt{d_*}}{q_*LdT^{1/2}} + \frac{16(4L+2\gamma_1)\sqrt{d_*}\sigma_s^2}{q_*BT^{1/2}}\right.$$
$$\left. + \frac{4\left((4-B)L + (B+2)\gamma_1\right)\sqrt{d_*}}{q_*BT^{1/2}} + \frac{16(2L+\gamma_1)\Xi C^2\sqrt{d_*}}{q_*BT^{1/2}} + \frac{16(2L+\gamma_1)\sigma_{dp}^2 d_m}{q_*T^{1/2}\sqrt{d_*}}\right\}, \tag{23}$$

*where $\Xi = 2\sqrt{2\pi}\exp(-\frac{C_0^2}{8\ell^2})$.*

*Proof.* We start from the result of an intermediate Lemma D.2, which quantifies the descent of the Lyapunov function, and we relegate the proof to Appendix D.

$$\mathbb{E}\left[V^{t+1} - V^t\right] \leq -\frac{1-P}{8}\min_m\{q_m\eta_m\}\mathbb{E}\left[\|\nabla_{\boldsymbol{w}}F(\boldsymbol{w}^t)\|^2\right] + \mathcal{A}_1 + \mathcal{A}_2 \qquad (24)$$

Note that $\mathcal{A}_1$ and $\mathcal{A}_2$ are constant terms defined in equation 35 and equation 39.

We first re-arranging the terms in equation 24, and take average over $0, 1, \ldots, T-1$:

$$\frac{1-P}{8T}\min_m\{q_m\eta_m\}\sum_{t=0}^{T-1}\mathbb{E}\left[\|\nabla_{\boldsymbol{w}}F(\boldsymbol{w}^t)\|^2\right]$$

$$\leq \frac{1}{T}\mathbb{E}\left[V^0 - V^T\right] + \mathcal{A}_1 + \mathcal{A}_2$$

$$\leq \frac{1}{T}\mathbb{E}\left[F^0 - F^T\right] + \mathcal{A}_1 + \mathcal{A}_2,$$

where the second inequality follows from the definition of $V^t$ in equation 22.

Dividing $\alpha = \frac{1}{8}\min_m\{q_m\eta_m\}$ from both sides, and plugging in $\mathcal{A}_1$ and $\mathcal{A}_2$, we have:

$$\frac{1-P}{T}\sum_{t=0}^{T-1}\mathbb{E}\left[\|\nabla_{\boldsymbol{w}}F(\boldsymbol{w}^t)\|^2\right]$$

$$\leq \frac{1}{\alpha T}\mathbb{E}\left[F^0 - F^T\right] + \frac{1}{\alpha}\sum_{m=0}^{M}q_m\eta_m\frac{L^2}{8}\lambda^2 d_m^3$$

$$+ \frac{1}{\alpha}\sum_{m=0}^{M}q_m\eta_m^2 L\left(\frac{4}{B}d_m\sigma_s^2 + \frac{(4-B)L^2}{4B}\lambda^2 d_m^3 + \frac{2P}{B}C^2 d_m + 2\sigma_{dp}^2 d_m\right)$$

$$+ \frac{1}{\alpha}L\lambda^2 d + \frac{1}{\alpha}\sum_{m=1}^{M}q_m\eta_m^2\gamma_1\left(\frac{L^2}{4}\lambda^2 d_m^3 + \frac{2}{B}d_m\sigma_s^2 + \frac{L^2}{2B}\lambda^2 d_m^3 + \frac{P}{B}C^2 d_m + \sigma_{dp}^2 d_m\right)$$

For the simplicity of analysis, we let $\eta_0 = \eta_m = \eta$, $q_* = \min_m q_m$, then $\alpha = \frac{\eta}{8}q_*$. Let $d_* = \max_m d_c$, and $\lambda \leq \frac{1}{Ld\sqrt{T}}$. Thus,

$$\frac{1-P}{T}\sum_{t=0}^{T-1}\mathbb{E}\left[\|\nabla_{\boldsymbol{w}}F(\boldsymbol{w}^t)\|^2\right]$$

$$\leq \frac{8}{q_*\eta T}\mathbb{E}\left[F^0 - F^T\right] + \frac{2d_*^3/d^2}{q_*T} + \frac{16\eta L}{q_*}\left(\frac{4}{B}d_*\sigma_s^2 + \frac{(4-B)d_*^3/d^2}{4BT} + \frac{2P}{B}C^2 d_* + 2\sigma_{dp}^2 d_*\right)$$

$$+ \frac{8}{q_*Ld\eta T} + \frac{16\eta\gamma_1}{q_*}\left(\frac{2}{B}d_*\sigma_s^2 + \frac{(B+2)d_*^3/d^2}{4BT} + \frac{P}{B}C^2 d_* + \sigma_{dp}^2 d_m\right)$$

$$\leq \frac{8}{q_*\eta T}\mathbb{E}\left[F^0 - F^T\right] + \frac{2d_*}{q_*T} + \frac{8}{q_*Ld\eta T} + \frac{16\eta(4L+2\gamma_1)d_*\sigma_s^2}{q_*B} + \frac{4\eta\left((4-B)L+(B+2)\gamma_1\right)d_*}{q_*BT}$$

$$+ \frac{16\eta(2L+\gamma_1)PC^2 d_*}{q_*B} + \frac{16\eta(2L+\gamma_1)\sigma_{dp}^2 d_*}{q_*}$$

where in the last step, we use the fact that $d > d_*$.

If we choose $\eta = \frac{1}{\sqrt{Td_*}}$, we can get the convergence rate:

$$\frac{1}{T}\sum_{t=0}^{T-1}\mathbb{E}\left[\|\nabla_{\boldsymbol{w}}F(\boldsymbol{w}^t)\|^2\right]$$

$$\leq \frac{1}{1-P}\left\{\frac{8\sqrt{d_*}}{q_*T^{1/2}}\mathbb{E}\left[F^0 - F^T\right] + \frac{2d_*}{q_*T} + \frac{8\sqrt{d_*}}{q_*LdT^{1/2}} + \frac{16(4L+2\gamma_1)\sqrt{d_*}\sigma_s^2}{q_*BT^{1/2}}\right.$$

$$\left. + \frac{4\left((4-B)L+(B+2)\gamma_1\right)\sqrt{d_*}}{q_*BT^{1/2}} + \frac{16(2L+\gamma_1)PC^2\sqrt{d_*}}{q_*BT^{1/2}} + \frac{16(2L+\gamma_1)\sigma_{dp}^2\sqrt{d_*}}{q_*T^{1/2}}\right\}$$

$$\leq \frac{1}{1-\Xi} \left\{ \frac{8\sqrt{d_*}}{q_* T^{1/2}} \mathbb{E}\left[F^0 - F^T\right] + \frac{2d_*}{q_* T} + \frac{8\sqrt{d_*}}{q_* L d T^{1/2}} + \frac{16(4L + 2\gamma_1)\sqrt{d_*}\sigma_s^2}{q_* B T^{1/2}} \right.$$

$$\left. + \frac{4\left((4-B)L + (B+2)\gamma_1\right)\sqrt{d_*}}{q_* B T^{1/2}} + \frac{16(2L + \gamma_1)\Xi C^2\sqrt{d_*}}{q_* B T^{1/2}} + \frac{16(2L + \gamma_1)\sigma_{dp}^2\sqrt{d_*}}{q_* T^{1/2}} \right\}$$

where the last step follows by Lemma A.8.

We thus conclude that the convergence rate is

$$\mathcal{O}\left(\frac{d_*^{1/2}\mathbb{E}\left[F^0 - F^T\right] + d_*^{1/2}\sigma_s^2 + d_*^{1/2}\sigma_{dp}^2}{T^{1/2}}\right),$$

where constants before terms have been omitted for simplicity. We note that our convergence rate is of the same order compared to centralized ZO optimization under the same smoothness assumption, up to an error term $\sigma_{dp}$ introduced by Differential Privacy. $\qquad\square$

## C  PROOF OF DIFFERENTIAL PRIVACY(THEOREM 5.1)

In this section, we give rigorous proof for the differential privacy guarantee in theorem 5.1.

We first introduce the definition of Gaussian differential privacy (GDP) (Dong et al., 2022) which will be useful in the proof for theorem 5.1. Compared with traditional DP defined in equation 4, this notion of privacy provides a much tighter composition theorem.

**Definition C.1** (Gaussian Differential Privacy). Let $G_\mu := T(\mathcal{N}(0,1), \mathcal{N}(\mu,1))$ for $\mu \geq 0$, where the trade-off function $T(P,Q) : [0,1] \to [0,1]$ is defined as $T(P,Q)(\alpha) = \inf(\beta_\phi : \alpha_\phi < \alpha)$. A mechanism $M$ is said to satisfy $\mu$-Gaussian Differential Privacy if it satisfies

$$T(M(X), M(X')) \geq G_\mu$$

For all neighboring dataset $X, X' \in \mathcal{X}^n$.

We construct our DP algorithm based on the Gaussian Mechanism. The Gaussian Mechanism of GDP is given by the following theorem.

**Theorem C.2** (Gaussian Mechanism for Gaussian Differential Privacy). *Define the Gaussian mechanism that operates on a statistic $\theta$ as $M(\theta) = \theta(X) + \sigma$, where $\sigma \sim \mathcal{N}(0, r^2 C_\theta^2/\mu^2)$, $r$ is the sample rate for a single datum, and $C_\theta$ is the $L_2$ sensitivity of $X$. Then, $M$ is $\mu$-GDP.*

The iterative nature of gradient-like algorithms calls for the composition theorem.

**Theorem C.3** (Composition of Gaussian Differential Privacy). *The $T$-fold composition of $\mu$-GDP mechanisms is $\sqrt{T}\mu$-GDP*

For the ease of comparison, we convert GDP to the common $(\epsilon, \delta)$-DP based on the following lossless conversion:

**Theorem C.4** (Conversion from Gaussian Differential Privacy to $(\epsilon, \delta)$-Differential Privacy). *A mechanism is $\mu$-GDP iff it is $(\epsilon, \delta(\epsilon))$-DP for all $\epsilon \geq 0$, where*

$$\delta(\epsilon) = \Phi\left(-\frac{\epsilon}{\mu} + \frac{\mu}{2}\right) - e^\epsilon \Phi\left(-\frac{\epsilon}{\mu} - \frac{\mu}{2}\right)$$

The proof of Theorem C.2-C.4 is given in (Dong et al., 2022).

We are now ready to present the proof for Theorem 5.1.

**Proof of Theorem 5.1**

*Proof.* First recall the definition of $\Delta_m^t$ defined in equation 8:

$$\Delta_m^t = \frac{1}{B} \sum_{i \in \mathcal{I}_m^{t_m}} \mathrm{clip}_C(\delta_{m,i}^{t,t_m}) + z_m^t$$

For a pair of neighboring dataset $X, X'$ differing in only one entry of data, the $L_2$ sensitivity $C_L$ of $\frac{1}{B} \sum_{i \in \mathcal{I}_m^{t_m}} \text{clip}_C(\delta_{m,i}^{t,t_m})$ follows by

$$C_L = \left\| \frac{1}{B} \sum_{i \in \mathcal{I}_m^{t_m}} \text{clip}_C(\delta_{m,i}^{t,t_m}) \right\|_2 \leq \frac{1}{B} \left\| \text{clip}_C(\delta_{m,i}^{t,t_m}) \right\|_2 \leq \frac{2C}{B}$$

The sample rate $r$ of a single data is seen to be the batch size $B$ divided by the size of the dataset $D$:

$$r = \frac{B}{D}$$

Note that in theorem 5.1, the standard variance of $z_m^t$ is given by

$$\sigma_{dp} = \frac{2C\sqrt{T}}{D\mu} = \frac{(B/D)(2C/B)\sqrt{T}}{\mu} = \frac{rC_L}{(\mu/\sqrt{T})}$$

By theorem C.2, the mechanism conducted in equation 8 satisfies $(\mu/\sqrt{T})$-Gaussian Differential Privacy. Further applying the composition of GDP in theorem C.3, and by the post-processing (Dwork et al., 2014) of differential privacy, we have that the whole training process of Algorithm 1 is $\mu$-GDP. We complete the proof by converting $\mu$-GDP to $(\epsilon, \delta)$-DP according to theorem C.4. $\qquad \square$

**Justification for Corollory 5.2** Based on Theorem 5.1, at time $t$, the model parameter on client $m$ is differentially private w.r.t. label information. Additionally, the input data $\xi_i$ does not contain any label information. Thus, the forward embedding is DP w.r.t labels.

# D INTERMEDIATE LEMMAS

**Lemma D.1** (Model Update With Delay). *Under Assumption A.1 to A.4, we have the following lemma:*

$$\mathbb{E}\left[F(\boldsymbol{w}^{t+1}) - F(\boldsymbol{w}^t)\right] \leq - \sum_{m=0}^{M} q_m \eta_m (1-P) \left(\frac{1}{4} - \frac{\eta_m L(B + 8d_m)}{2B}\right) \mathbb{E}\left[\|\nabla_{w_m} F(\boldsymbol{w}^t)\|^2\right]$$

$$+ \sum_{m=0}^{M} q_m \eta_m L \left(\frac{1}{2} + 2\eta_m L^2\right) \mathbb{E}\left[\|\tilde{\chi}^t - \chi^t\|^2\right] + \mathcal{A}_1 \qquad (25)$$

*Proof.* By assumption A.2:

$$F_\lambda(\boldsymbol{w}^{t+1}) \leq F_\lambda(\boldsymbol{w}^t) + \langle \nabla_{\boldsymbol{w}} F_\lambda(\boldsymbol{w}^t), \boldsymbol{w}^{t+1} - \boldsymbol{w}^t \rangle + \frac{L}{2} \|\boldsymbol{w}^{t+1} - \boldsymbol{w}^t\|^2$$

$$= F_\lambda(\boldsymbol{w}^t) - \eta_0 \langle \nabla_{w_0} F_\lambda(\boldsymbol{w}^t), \breve{G}_0^t(\tilde{\boldsymbol{w}}^t) \rangle + \frac{L\eta_0^2}{2} \left\| \breve{G}_0^t(\tilde{\boldsymbol{w}}^t) \right\|^2$$

$$- \eta_m \langle \nabla_{w_m} F_\lambda(\boldsymbol{w}^t), \breve{G}_m^t(\tilde{\boldsymbol{w}}^t) \rangle + \frac{L\eta_m^2}{2} \left\| \breve{G}_m^t(\tilde{\boldsymbol{w}}^t) \right\|^2$$

$$\mathbb{E}\left[F_\lambda(\boldsymbol{w}^{t+1})\right] \leq \mathbb{E}\left[F_\lambda(\boldsymbol{w}^t)\right] \underbrace{- \eta_0 \mathbb{E}\langle \nabla_{w_0} F_\lambda(\boldsymbol{w}^t), \breve{G}_0^t(\tilde{\boldsymbol{w}}^t) \rangle}_{\mathcal{E}_1} + \underbrace{\frac{L\eta_0^2}{2} \mathbb{E}\left[\left\| \breve{G}_0^t(\tilde{\boldsymbol{w}}^t) \right\|^2\right]}_{\mathcal{E}_2}$$

$$\underbrace{- \eta_{m_k} \mathbb{E}\langle \nabla_{w_{m_k}} F_\lambda(\boldsymbol{w}^t), \breve{G}_{m_k}^t(\tilde{\boldsymbol{w}}^t) \rangle}_{\mathcal{E}_3} + \underbrace{\frac{L\eta_{m_k}^2}{2} \mathbb{E}\left[\left\| \breve{G}_{m_k}^t(\tilde{\boldsymbol{w}}^t) \right\|^2\right]}_{\mathcal{E}_4}$$

$$(26)$$

where in the second step we take expectation on both sides, first w.r.t. the random direction $u$, DP noise $z$, and the clipping event $Q$, then w.r.t the client $m_k$. We bound $\mathcal{E}_1$ as the following:

$$- \eta_0 \mathbb{E}\langle \nabla_{w_0} F_\lambda(\boldsymbol{w}^t), \breve{G}_0^t(\tilde{\boldsymbol{w}}^t) \rangle$$

$$= -\eta_0 \mathbb{E}\langle \nabla_{w_0} F_\lambda(\boldsymbol{w}^t), \breve{G}_0^t(\tilde{\boldsymbol{w}}^t) - (1-P)\nabla_{w_0} F_\lambda(\tilde{\boldsymbol{w}}^t) + (1-P)\nabla_{w_0} F_\lambda(\tilde{\boldsymbol{w}}^t)\rangle$$

$$= -\eta_0 \mathbb{E}\langle \nabla_{w_0} F_\lambda(\boldsymbol{w}^t), \breve{G}_0^t(\tilde{\boldsymbol{w}}^t) - (1-P)\nabla_{w_0} F_\lambda(\tilde{\boldsymbol{w}}^t)\rangle$$

$$-\eta_0 \mathbb{E}\langle \nabla_{w_0} F_\lambda(\boldsymbol{w}^t), (1-P)\nabla_{w_0} F_\lambda(\tilde{\boldsymbol{w}}^t) - (1-P)\nabla_{w_0} F_\lambda(\boldsymbol{w}^t) + (1-P)\nabla_{w_0} F_\lambda(\boldsymbol{w}^t)\rangle$$

$$\overset{1)}{=} -(1-P)\eta_0 \mathbb{E}\langle \nabla_{w_0} F_\lambda(\boldsymbol{w}^t), \nabla_{w_0} F_\lambda(\tilde{\boldsymbol{w}}^t) - \nabla_{w_0} F_\lambda(\boldsymbol{w}^t)\rangle$$

$$-\eta_0 \mathbb{E}\langle \nabla_{w_0} F_\lambda(\boldsymbol{w}^t), (1-P)\nabla_{w_0} F_\lambda(\boldsymbol{w}^t)\rangle$$

$$\overset{2)}{\le} \frac{(1-P)\eta_0}{2}\mathbb{E}\left[\|\nabla_{w_0} F_\lambda(\boldsymbol{w}^t)\|^2\right] + \frac{(1-P)\eta_0}{2}\mathbb{E}\left[\|\nabla_{w_0} F_\lambda(\tilde{\boldsymbol{w}}^t) - \nabla_{w_0} F_\lambda(\boldsymbol{w}^t)\|^2\right] - (1-P)\eta_0\mathbb{E}\left[\|\nabla_{w_0} F_\lambda(\boldsymbol{w}^t)\|^2\right]$$

$$\overset{3)}{\le} -\frac{(1-P)\eta_0}{2}\mathbb{E}\left[\|\nabla_{w_0} F_\lambda(\boldsymbol{w}^t)\|^2\right] + \frac{\eta_0 L}{2}\mathbb{E}\left[\|\tilde{\chi}^t - \chi^t\|^2\right] \tag{27}$$

where in 1) we use the fact that $\mathbb{E}\left[\breve{G}_0^t(\tilde{\boldsymbol{w}}^t) - (1-P)\nabla_{x_0} F_\lambda(\tilde{\boldsymbol{w}}^t)\right] = 0$, in 2) we applied the Cauchy–Schwarz inequality, and 3) follows by the smoothness of $F_\lambda$ and the fact that $1 - P \le 1$

For $\mathcal{E}_2$, we can further bound it based on Assumption A.2:

$$\frac{1}{2}\mathbb{E}\left[\left\|\breve{G}_0^t(\tilde{\boldsymbol{w}}^t)\right\|^2\right]$$

$$= \frac{1}{2}\mathbb{E}\left[\left\|\breve{G}_0^t(\tilde{\boldsymbol{w}}^t) - (1-P)\nabla_{x_0} F_\lambda(\boldsymbol{w}^t) + (1-P)\nabla_{x_0} F_\lambda(\boldsymbol{w}^t)\right\|^2\right]$$

$$\overset{1)}{\le} \mathbb{E}\left[\left\|\breve{G}_0^t(\tilde{\boldsymbol{w}}^t) - (1-P)\nabla_{x_0} F_\lambda(\boldsymbol{w}^t)\right\|^2\right] + (1-P)^2\mathbb{E}\left[\|\nabla_{x_0} F_\lambda(\boldsymbol{w}^t)\|^2\right]$$

$$= \mathbb{E}\left[\left\|\breve{G}_0^t(\tilde{\boldsymbol{w}}^t) - (1-P)\nabla_{x_0} F_\lambda(\tilde{\boldsymbol{w}}^t) + (1-P)\nabla_{x_0} F_\lambda(\tilde{\boldsymbol{w}}^t) - (1-P)\nabla_{x_0} F_\lambda(\boldsymbol{w}^t)\right\|^2\right]$$

$$+ (1-P)^2\mathbb{E}\left[\|\nabla_{x_0} F_\lambda(\boldsymbol{w}^t)\|^2\right]$$

$$\overset{2)}{\le} 2\mathbb{E}\left[\left\|\breve{G}_0^t(\tilde{\boldsymbol{w}}^t) - (1-P)\nabla_{x_0} F_\lambda(\tilde{\boldsymbol{w}}^t)\right\|^2\right] + 2(1-P)^2\mathbb{E}\left[\|\nabla_{x_0} F_\lambda(\tilde{\boldsymbol{w}}^t) - \nabla_{x_0} F_\lambda(\boldsymbol{w}^t)\|^2\right]$$

$$+ (1-P)^2\mathbb{E}\left[\|\nabla_{x_0} F_\lambda(\boldsymbol{w}^t)\|^2\right]$$

$$\overset{3)}{\le} \frac{2(1-P)}{B}\left(2d_0\mathbb{E}\left[\|\nabla_{x_0} F(\boldsymbol{w}^t)\|^2\right] + 2d_0\sigma_s^2 + \frac{L^2}{2}\lambda^2 d_0^3\right) + \frac{2P}{B}C^2 d_0$$

$$- \frac{2(1-P)^2}{B}\mathbb{E}\left[\|\nabla_{x_0} F_\lambda(\boldsymbol{w}^t)\|^2\right] + 2\sigma_{dp}^2 d_0 + 2(1-P)^2 L^2\mathbb{E}\left[\|\tilde{\chi}^t - \chi^t\|^2\right] + (1-P)^2\mathbb{E}\left[\|\nabla_{x_0} F_\lambda(\boldsymbol{w}^t)\|^2\right]$$

$$\overset{4)}{\le} \frac{4(1-P)d_0}{B}\mathbb{E}\left[\|\nabla_{x_0} F(\boldsymbol{w}^t)\|^2\right] + (1-P)\mathbb{E}\left[\|\nabla_{x_0} F_\lambda(\boldsymbol{w}^t)\|^2\right] + 2L^2\mathbb{E}\left[\|\tilde{\chi}^t - \chi^t\|^2\right] + \mathcal{G}_0 \tag{28}$$

where in 1) and 2) we applied the Cauchy–Schwarz inequality, and in 3) we substitute equation 21 in and use the $L$-smoothness of $F_\lambda$, and in $(iv)$ we use the fact that $1 - P \le 1$ and let

$$\mathcal{G}_0 = \frac{4}{B}d_0\sigma_s^2 + \frac{L^2}{B}\lambda^2 d_0^3 + \frac{2P}{B}C^2 d_0 + 2\sigma_{dp}^2 d_0$$

Similarly, For $\mathcal{E}_3$:

$$-\eta_{m_k}\mathbb{E}\langle \nabla_{w_{m_k}} F_\lambda(\boldsymbol{w}^t), \breve{G}_m^t(\tilde{\boldsymbol{w}}^t)\rangle \le -\frac{(1-P)\eta_{m_k}}{2}\mathbb{E}\left[\left\|\nabla_{w_{m_k}} F_\lambda(\boldsymbol{w}^t)\right\|^2\right] + \frac{\eta_{m_k} L}{2}\mathbb{E}\left[\|\tilde{\chi}^t - \chi^t\|^2\right] \tag{29}$$

And for $\mathcal{E}_4$:

$$\frac{1}{2}\mathbb{E}\left[\left\|\breve{G}_{m_k}^t(\tilde{\boldsymbol{w}}^t)\right\|^2\right] \le \frac{4(1-P)d_m}{B}\mathbb{E}\left[\left\|\nabla_{w_{m_k}} F(\boldsymbol{w}^t)\right\|^2\right] + (1-P)\mathbb{E}\left[\left\|\nabla_{w_{m_k}} F_\lambda(\boldsymbol{w}^t)\right\|^2\right] \tag{30}$$

$$+ 2L^2\mathbb{E}\left[\|\tilde{\chi}^t - \chi^t\|^2\right] + \mathcal{G}_m \tag{31}$$

where we let

$$\mathcal{G}_m = \frac{4}{B} d_m \sigma_s^2 + \frac{L^2}{B} \lambda^2 d_m^3 + \frac{2P}{B} C^2 d_m + 2\sigma_{dp}^2 d_m \tag{32}$$

Substituting equation 27, equation 28, equation 29, and equation 31 into equation 26, we have

$$\mathbb{E}\left[F(\boldsymbol{w}^{t+1}) - F(\boldsymbol{w}^t)\right]$$

$$\leq \mathbb{E}\left[F_\lambda(\boldsymbol{w}^t)\right] - \frac{(1-P)\eta_0}{2} \mathbb{E}\left[\|\nabla_{w_0} F_\lambda(\boldsymbol{w}^t)\|^2\right] + \frac{\eta_0 L}{2} \mathbb{E}\left[\|\tilde{\chi}^t - \chi^t\|^2\right]$$

$$+ \frac{4(1-P)d_0 L \eta_0^2}{B} \mathbb{E}\left[\|\nabla_{w_0} F(\boldsymbol{w}^t)\|^2\right] + (1-P)L\eta_0^2 \mathbb{E}\left[\|\nabla_{w_0} F_\lambda(\boldsymbol{w}^t)\|^2\right] + 2L^3 \eta_0^2 \mathbb{E}\left[\|\tilde{\chi}^t - \chi^t\|^2\right] + L\eta_0^2 \mathcal{G}_0$$

$$- \frac{(1-P)\eta_{m_k}}{2} \mathbb{E}\left[\left\|\nabla_{w_{m_k}} F_\lambda(\boldsymbol{w}^t)\right\|^2\right] + \frac{\eta_{m_k} L}{2} \mathbb{E}\left[\|\tilde{\chi}^t - \chi^t\|^2\right]$$

$$+ \frac{4(1-P)d_{m_k} L \eta_{m_k}^2}{B} \mathbb{E}\left[\left\|\nabla_{w_{m_k}} F(\boldsymbol{w}^t)\right\|^2\right] + (1-P)L\eta_{m_k}^2 \mathbb{E}\left[\left\|\nabla_{w_{m_k}} F_\lambda(\boldsymbol{w}^t)\right\|^2\right]$$

$$+ 2L^3 \eta_{m_k}^2 \mathbb{E}\left[\|\tilde{\chi}^t - \chi^t\|^2\right] + L\eta_{m_k}^2 \mathcal{G}_m$$

$$\leq \mathbb{E}\left[F_\lambda(\boldsymbol{w}^t)\right] - \eta_0(1-P)\left(\frac{1}{2} - \eta_0 L\right) \mathbb{E}\left[\|\nabla_{w_0} F_\lambda(\boldsymbol{w}^t)\|^2\right] + \eta_0 L\left(\frac{1}{2} + 2\eta_0 L^2\right) \mathbb{E}\left[\|\tilde{\chi}^t - \chi^t\|^2\right]$$

$$+ \eta_0^2 \frac{4(1-P)d_0 L}{B} \mathbb{E}\left[\|\nabla_{w_0} F(\boldsymbol{w}^t)\|^2\right] + \eta_0^2 L \mathcal{G}_0$$

$$- \sum_{m=1}^{M} q_m \eta_m (1-P)\left(\frac{1}{2} - \eta_m L\right) \mathbb{E}\left[\|\nabla_{w_m} F_\lambda(\boldsymbol{w}^t)\|^2\right] + \sum_{m=1}^{M} q_m \eta_m L\left(\frac{1}{2} + 2\eta_m L^2\right) \mathbb{E}\left[\|\tilde{\chi}^t - \chi^t\|^2\right]$$

$$+ \sum_{m=1}^{M} q_m \eta_m^2 \frac{4(1-P)d_m L}{B} \mathbb{E}\left[\|\nabla_{w_m} F(\boldsymbol{w}^t)\|^2\right] + \sum_{m=1}^{M} q_m \eta_m^2 L \mathcal{G}_m \tag{33}$$

where in the last inequality we further take expectation w.r.t. client $M$ and combine similar terms.

From equation 33, we utilize the properties of the smooth function equation 14 and equation 17 to turn all the smooth function $F_\lambda$ into the true loss function $F$:

$$\mathbb{E}\left[F(\boldsymbol{w}^{t+1}) - F(\boldsymbol{w}^t)\right] \overset{1)}{\leq} \mathbb{E}\left[F_\lambda(\boldsymbol{w}^{t+1}) - F_\lambda(\boldsymbol{w}^t)\right] + L\lambda^2 d$$

$$\overset{2)}{\leq} -\eta_0(1-P)\left(\frac{1}{2} - \eta_0 L\right)\left(\frac{1}{2}\mathbb{E}\left[\|\nabla_{x_0} F(\boldsymbol{w}^t)\|^2\right] - \frac{L^2}{4}\lambda^2 d_0^3\right) + \eta_0 L\left(\frac{1}{2} + 2\eta_0 L^2\right) \mathbb{E}\left[\|\tilde{\chi}^t - \chi^t\|^2\right]$$

$$+ \eta_0^2 \frac{4(1-P)d_0 L}{B} \mathbb{E}\left[\|\nabla_{x_0} F(\boldsymbol{w}^t)\|^2\right] + \eta_0^2 L \mathcal{G}_0$$

$$- \sum_{m=1}^{M} q_m \eta_m (1-P)\left(\frac{1}{2} - \eta_m L\right)\left(\frac{1}{2}\mathbb{E}\left[\|\nabla_{w_m} F(\boldsymbol{w}^t)\|^2\right] - \frac{L^2}{4}\lambda^2 d_m^3\right)$$

$$+ \sum_{m=1}^{M} q_m \eta_m L\left(\frac{1}{2} + 2\eta_m L^2\right) \mathbb{E}\left[\|\tilde{\chi}^t - \chi^t\|^2\right]$$

$$+ \sum_{m=1}^{M} q_m \eta_m^2 \frac{4(1-P)d_m L}{B} \mathbb{E}\left[\|\nabla_{w_m} F(\boldsymbol{w}^t)\|^2\right] + \sum_{m=1}^{M} q_m \eta_m^2 L \mathcal{G}_m + L\lambda^2 d$$

$$\overset{3)}{\leq} -\sum_{m=0}^{M} q_m \eta_m (1-P)\left(\frac{1}{4} - \frac{\eta_m L(B + 8d_m)}{2B}\right) \mathbb{E}\left[\|\nabla_{w_m} F(\boldsymbol{w}^t)\|^2\right]$$

$$+ \sum_{m=0}^{M} q_m \eta_m L\left(\frac{1}{2} + 2\eta_m L^2\right) \mathbb{E}\left[\|\tilde{\chi}^t - \chi^t\|^2\right] + \sum_{m=0}^{M} q_m \eta_m \left(\frac{1}{2} - \eta_m L\right)\frac{L^2}{4}\lambda^2 d_m^3$$

$$+ \sum_{m=0}^{M} q_m \eta_m^2 L \mathcal{G}_m + L\lambda^2 d$$

$$\overset{4)}{\leq} -\sum_{m=0}^{M} q_m \eta_m (1-P) \left( \frac{1}{4} - \frac{\eta_m L(B + 8d_m)}{2B} \right) \mathbb{E}\left[ \|\nabla_{w_m} F(\boldsymbol{w}^t)\|^2 \right]$$

$$+ \sum_{m=0}^{M} q_m \eta_m L \left( \frac{1}{2} + 2\eta_m L^2 \right) \mathbb{E}\left[ \|\tilde{\chi}^t - \chi^t\|^2 \right]$$

$$+ \sum_{m=0}^{M} q_m \eta_m \frac{L^2}{8} \lambda^2 d_m^3 + \sum_{m=0}^{M} q_m \eta_m^2 L \left( \frac{4}{B} d_m \sigma_s^2 + \frac{(4-B)L^2}{4B} \lambda^2 d_m^3 + \frac{2P}{B} C^2 d_m + 2\sigma_{dp}^2 d_m \right) + L\lambda^2 d$$

$$\overset{5)}{\leq} -\sum_{m=0}^{M} q_m \eta_m (1-P) \left( \frac{1}{4} - \frac{\eta_m L(B + 8d_m)}{2B} \right) \mathbb{E}\left[ \|\nabla_{w_m} F(\boldsymbol{w}^t)\|^2 \right]$$

$$+ \sum_{m=0}^{M} q_m \eta_m L \left( \frac{1}{2} + 2\eta_m L^2 \right) \mathbb{E}\left[ \|\tilde{\chi}^t - \chi^t\|^2 \right] + \mathcal{A}_1 \tag{34}$$

where 1) and 2) follows from equation equation 14 and equation 17 in Lemma A.6 respectively. In 3), we let $q_0 = 1$ and combine similar terms. In 4), we substitute in equation 32. Lastly, in 5), we denote

$$\mathcal{A}_1 = \sum_{m=0}^{M} q_m \eta_m \frac{L^2}{8} \lambda^2 d_m^3 + \sum_{m=0}^{M} q_m \eta_m^2 L \left( \frac{4}{B} d_m \sigma_s^2 + \frac{(4-B)L^2}{4B} \lambda^2 d_m^3 + \frac{2P}{B} C^2 d_m + 2\sigma_{dp}^2 d_m \right) + L\lambda^2 d \tag{35}$$

for the convenience of notation.

We thus complete the proof. $\qquad\square$

Now, recall the definition of the Lyapunov function $V^t$:

$$V^t = F(\boldsymbol{w}^t) + \sum_{i=1}^{\tau} \gamma_i \|\chi^{t+1-i} - \chi^{t-i}\|^2$$

We utilize the Lyapunov function to prove the following lemma for eliminating model delay.

**Lemma D.2.** *Under Assumption A.1-A.4, and assume the client delay $\tau_m$ is uniformly upper bounded by $\tau$, we have the following lemma:*

$$\mathbb{E}\left[ V^{t+1} - V^t \right] \leq -\frac{1-P}{8} \min_m \{q_m \eta_m\} \mathbb{E}\left[ \|\nabla_{\boldsymbol{w}} F(\boldsymbol{w}^t)\|^2 \right] + \mathcal{A}_1 + \mathcal{A}_2$$

*Proof.* Before we give the proof of Lemma D.2, we first provide some useful facts that reveal properties of the delayed parameters.

Recall that $\tilde{\chi}^t$ denote the delayed parameters on all clients, and $\chi^t$ denote the non-delayed version. Let $\mathcal{F}_1 = \mathbb{E}\left[ \|\chi^{t+1} - \chi^t\|^2 \right]$, $\mathcal{F}_2 = \mathbb{E}\left[ \|\tilde{\chi}^t - \chi^t\|^2 \right]$.

For $\mathcal{F}_1$:

$$\mathbb{E}\left[ \|\chi^{t+1} - \chi^t\|^2 \right]$$

$$= \eta_m^2 \mathbb{E}\left[ \left\| \breve{G}_{m_k}^t(\tilde{\boldsymbol{w}}^t) \right\|^2 \right]$$

$$\leq \sum_{m=1}^{M} q_m \eta_m^2 \frac{2(1-P)d_m}{B} \mathbb{E}\left[ \left\| \nabla_{w_{m_k}} F(\boldsymbol{w}^t) \right\|^2 \right] + \sum_{m=1}^{M} q_m \eta_m^2 \frac{(1-P)}{2} \mathbb{E}\left[ \left\| \nabla_{w_{m_k}} F_\lambda(\boldsymbol{w}^t) \right\|^2 \right]$$

$$+ \sum_{m=1}^{M} q_m \eta_m^2 \left( L^2 \mathbb{E}\left[ \|\tilde{\chi}^t - \chi^t\|^2 \right] + \frac{1}{2} \mathcal{G}_m \right)$$

$$\leq \sum_{m=1}^{M} q_m \eta_m^2 \frac{2(1-P)d_m}{B} \mathbb{E}\left[ \left\| \nabla_{w_{m_k}} F(\boldsymbol{w}^t) \right\|^2 \right] + \sum_{m=1}^{M} q_m \eta_m^2 \frac{(1-P)}{2} \left( 2\mathbb{E}\left[ \left\| \nabla_{w_{m_k}} F(\boldsymbol{w}^t) \right\|^2 \right] + \frac{L^2}{2} \lambda^2 d_m^3 \right)$$

$$+ \sum_{m=1}^{M} q_m \eta_m^2 \left( L^2 \mathbb{E}\left[\|\tilde{\chi}^t - \chi^t\|^2\right] + \frac{1}{2}\mathcal{G}_m \right)$$

$$= \sum_{m=1}^{M} q_m \eta_m^2 \frac{(1-P)(2d_m+B)}{B} \mathbb{E}\left[\left\|\nabla_{w_{m_k}} F(\boldsymbol{w}^t)\right\|^2\right] + \sum_{m=1}^{M} q_m \eta_m^2 \left( \frac{L^2}{4}\lambda^2 d_m^3 + L^2 \mathbb{E}\left[\|\tilde{\chi}^t - \chi^t\|^2\right] + \frac{1}{2}\mathcal{G}_m \right)$$

$$(36)$$

For $\mathcal{F}_2$, under uniformly bounded delay, we have

$$\mathbb{E}\left[\|\tilde{\chi}^t - \chi^t\|^2\right]$$

$$\leq \mathbb{E}\left[\left\|\sum_{i=1}^{\tau}(\chi^{i+1} - \chi^i)\right\|^2\right]$$

$$\leq \tau \sum_{i=1}^{\tau} \mathbb{E}\left[\left\|\chi^{i+1} - \chi^i\right\|^2\right] \tag{37}$$

where the last inequality follows by Cauchy-Schwarz Inequality.
By the definition of $V^t$:

$$\mathbb{E}\left[V^{t+1} - V^t\right]$$

$$= \mathbb{E}\left[F(\boldsymbol{w}^{t+1}) + \sum_{i=1}^{\tau} \gamma_i \left\|\chi^{t+2-i} - \chi^{t+1-i}\right\|^2\right] - \mathbb{E}\left[F(\boldsymbol{w}^t) + \sum_{i=1}^{\tau} \gamma_i \left\|\chi^{t+1-i} - \chi^{t-i}\right\|^2\right]$$

$$= \mathbb{E}\left[F(\boldsymbol{w}^{t+1}) - F(\boldsymbol{w}^t)\right] + \sum_{i=1}^{\tau} \gamma_i \mathbb{E}\left[\left\|\chi^{t+2-i} - \chi^{t+1-i}\right\|^2\right] - \sum_{i=1}^{\tau} \gamma_i \mathbb{E}\left[\left\|\chi^{t+1-i} - \chi^{t-i}\right\|^2\right]$$

$$\overset{1)}{\leq} - \sum_{m=0}^{M} q_m \eta_m (1-P) \left( \frac{1}{4} - \frac{\eta_m L(B+8d_m)}{2B} \right) \mathbb{E}\left[\|\nabla_{w_m} F(\boldsymbol{w}^t)\|^2\right]$$

$$+ \sum_{m=0}^{M} q_m \eta_m L \left( \frac{1}{2} + 2\eta_m L^2 \right) \mathbb{E}\left[\|\tilde{\chi}^t - \chi^t\|^2\right] + \mathcal{A}_1$$

$$+ \gamma_1 \underbrace{\mathbb{E}\left[\left\|\chi^{t+1} - \chi^t\right\|^2\right]}_{\mathcal{F}_1} + \sum_{i=1}^{\tau-1}(\gamma_{i+1} - \gamma_i)\mathbb{E}\left[\left\|\chi^{t+1-i} - \chi^{t-i}\right\|^2\right] - \gamma_\tau \mathbb{E}\left[\left\|\chi^{t+1-\tau} - \chi^{t-\tau}\right\|^2\right]$$

$$\overset{2)}{\leq} - \sum_{m=0}^{M} q_m \eta_m (1-P) \left( \frac{1}{4} - \frac{\eta_m L(B+8d_m)}{2B} \right) \mathbb{E}\left[\|\nabla_{w_m} F(\boldsymbol{w}^t)\|^2\right]$$

$$+ \sum_{m=0}^{M} q_m \eta_m L \left( \frac{1}{2} + 2\eta_m L^2 \right) \mathbb{E}\left[\|\tilde{\chi}^t - \chi^t\|^2\right] + \mathcal{A}_1$$

$$+ \sum_{m=1}^{M} q_m \eta_m^2 \gamma_1 \frac{(1-P)(2d_m+B)}{B} \mathbb{E}\left[\left\|\nabla_{w_{m_k}} F(\boldsymbol{w}^t)\right\|^2\right] + \sum_{m=1}^{M} q_m \eta_m^2 \gamma_1 \left( \frac{L^2}{4}\lambda^2 d_m^3 + L^2 \mathbb{E}\left[\|\tilde{\chi}^t - \chi^t\|^2\right] + \frac{1}{2}\mathcal{G}_m \right)$$

$$+ \sum_{i=1}^{\tau-1}(\gamma_{i+1} - \gamma_i)\mathbb{E}\left[\left\|\chi^{t+1-i} - \chi^{t-i}\right\|^2\right] - \gamma_\tau \mathbb{E}\left[\left\|\chi^{t+1-\tau} - \chi^{t-\tau}\right\|^2\right]$$

$$\overset{3)}{=} - \eta_0(1-P) \left( \frac{1}{4} - \frac{\eta_0 L(B+8d_0)}{2B} \right) \mathbb{E}\left[\|\nabla_{x_0} F(\boldsymbol{w}^t)\|^2\right]$$

$$- \sum_{m=1}^{M} q_m \eta_m (1-P) \left( \frac{1}{4} - \frac{\eta_m L(B+8d_m)}{2B} - \frac{\eta_m \gamma_1(2d_m+B)}{B} \right) \mathbb{E}\left[\|\nabla_{w_m} F(\boldsymbol{w}^t)\|^2\right]$$

$$+ \left\{ \eta_0 L \left( \frac{1}{2} + 2\eta_0 L^2 \right) + \sum_{m=1}^{M} q_m \eta_m L \left( \frac{1}{2} + 2\eta_m L^2 + \eta_m \gamma_1 L \right) \right\} \underbrace{\mathbb{E}\left[\|\tilde{\chi}^t - \chi^t\|^2\right]}_{\mathcal{F}_2}$$

$$+ \mathcal{A}_1 + \sum_{m=1}^{M} q_m \eta_m^2 \gamma_1 \left( \frac{L^2}{4} \lambda^2 d_m^3 + \frac{1}{2} \mathcal{G}_m \right)$$

$$+ \sum_{i=1}^{\tau-1} (\gamma_{i+1} - \gamma_i) \mathbb{E} \left[ \left\| \chi^{t+1-i} - \chi^{t-i} \right\|^2 \right] - \gamma_\tau \mathbb{E} \left[ \left\| \chi^{t+1-\tau} - \chi^{t-\tau} \right\|^2 \right]$$

$$\overset{4)}{\leq} - \eta_0 (1 - P) \left( \frac{1}{4} - \frac{\eta_0 L(B + 8d_0)}{2B} \right) \mathbb{E} \left[ \left\| \nabla_{x_0} F(\boldsymbol{w}^t) \right\|^2 \right]$$

$$- \sum_{m=1}^{M} q_m \eta_m (1 - P) \left( \frac{1}{4} - \frac{\eta_m L(B + 8d_m)}{2B} - \frac{\eta_m \gamma_1 (2d_m + B)}{B} \right) \mathbb{E} \left[ \left\| \nabla_{w_m} F(\boldsymbol{w}^t) \right\|^2 \right]$$

$$+ \sum_{i=1}^{\tau-1} \left( \gamma_{i+1} - \gamma_i + \tau \left( \eta_0 L \left( \frac{1}{2} + 2\eta_0 L^2 \right) + \sum_{m=1}^{M} q_m \eta_m L \left( \frac{1}{2} + 2\eta_m L^2 + \eta_m \gamma_1 L \right) \right) \right) \mathbb{E} \left[ \left\| \chi^{t+1-i} - \chi^{t-i} \right\|^2 \right]$$

$$- \left( \gamma_\tau - \tau \left( \eta_0 L \left( \frac{1}{2} + 2\eta_0 L^2 \right) + \sum_{m=1}^{M} q_m \eta_m L \left( \frac{1}{2} + 2\eta_m L^2 + \eta_m \gamma_1 L \right) \right) \right) \mathbb{E} \left[ \left\| \chi^{t+1-\tau} - \chi^{t-\tau} \right\|^2 \right]$$

$$+ \mathcal{A}_1 + \mathcal{A}_2. \tag{38}$$

Above, we used Lemma D.1 in step $(1)$, substituted in equation 36 for $\mathcal{F}_1$ in step $(2)$, substituted in equation 37 for $\mathcal{F}_2$ in step $(3)$, and defined

$$\mathcal{A}_2 := \sum_{m=1}^{M} q_m \eta_m^2 \gamma_1 \left( \frac{L^2}{4} \lambda^2 d_m^3 + \frac{1}{2} \mathcal{G}_m \right)$$

$$= \sum_{m=1}^{M} q_m \eta_m^2 \gamma_1 \left( \frac{L^2}{4} \lambda^2 d_m^3 + \frac{2}{B} d_m \sigma_s^2 + \frac{L^2}{2B} \lambda^2 d_m^3 + \frac{P}{B} C^2 d_m + \sigma_{dp}^2 d_m \right). \tag{39}$$

From equation 38, we choose the following relationship for $\gamma_1, \gamma_2, \ldots, \gamma_m$:

$$\gamma_1 = \frac{\tau^2 \left( \eta_0 L \left( \frac{1}{2} + 2\eta_0 L^2 \right) + \sum_{m=1}^{M} q_m \eta_m L \left( \frac{1}{2} + 2\eta_m L^2 \right) \right)}{1 - \tau^2 \sum_{m=1}^{M} q_m \eta_m^2 L^2} \tag{40}$$

$$\gamma_{i+1} = \gamma_i - \tau \left( \eta_0 L \left( \frac{1}{2} + 2\eta_0 L^2 \right) + \sum_{m=1}^{M} q_m \eta_m L \left( \frac{1}{2} + 2\eta_m L^2 + \eta_m \gamma_1 L \right) \right)$$

and we can verify that

$$\gamma_\tau - \tau \left( \eta_0 L \left( \frac{1}{2} + 2\eta_0 L^2 \right) + \sum_{m=1}^{M} q_m \eta_m L \left( \frac{1}{2} + 2\eta_m L^2 + \eta_m \gamma_1 L \right) \right) \geq 0$$

We further let

$$\eta_0 \leq \frac{B}{4L(B + 8d_0)}, \eta_m \leq \frac{B}{4L(B + 8d_0) + 8\gamma_1 (2d_m + B)},$$

and we finally have

$$\mathbb{E} \left[ V^{t+1} - V^t \right] \leq -\frac{\eta_0}{8} (1 - P) \mathbb{E} \left[ \left\| \nabla_{x_0} F(\boldsymbol{w}^t) \right\|^2 \right] - \sum_{m=1}^{M} \frac{q_m \eta_m}{8} (1 - P) \mathbb{E} \left[ \left\| \nabla_{w_m} F(\boldsymbol{w}^t) \right\|^2 \right] + \mathcal{A}_1 + \mathcal{A}_2$$

$$\leq -\frac{1 - P}{8} \min_m q_m \eta_m \mathbb{E} \left[ \left\| \nabla_{\boldsymbol{w}} F(\boldsymbol{w}^t) \right\|^2 \right] + \mathcal{A}_1 + \mathcal{A}_2 \tag{41}$$

which completes the proof. $\qquad \square$

# E ADDITIONAL DETAILS ON EXPERIMENT

In this section, we would like to give a brief introduction of the datasets and model structures and further justify the choice of the experimental designs below. We follow the experimental settings of existing works(Chen et al., 2020a; Xie et al., 2024; Castiglia et al., 2023) for a fair comparison.

**Dataset** The choices of datasets cover a large range of tasks, including:

- MNIST and CIFAR-10: standard image benchmarks in machine learning.

- ModelNet40: multi-view 3D object classification dataset; each object has 12 views from different angles, which naturally lends itself to feature partitioning across clients in VFL.

- Amazon Reviews: sentiment analysis task, which we include to evaluate our algorithm in the NLP domain.

**Data partition and model selection** For MNIST, we use a two-layer CNN model, for VFL's data partitioning, we split the images by row evenly into 7 sub-images and assign them to 7 clients. For CIFAR-10 we use a four-layer CNN model, and partition each image into 2×2, 4 patches of the same size for 4 clients. For ModelNet40, we use a ResNet-18 model, and partition each object into 12 different camera views and allocate them to 12 clients. For Amazon Reviews, we use a pre-trained BERT (Devlin, 2018) model, and split the tokenized data input into 3 paragraphs of the same number of tokens and distributed them across 3 clients. For all four datasets, we use a fully connected model of two linear layers with ReLU activations as the server model.

# F    DISCUSSION OF DIFFERENTIAL PRIVACY UNDER VARIOUS SETTINGS

**Overview of DPZV's Privacy Mechanisms.** `DPZV` protects privacy through two main mechanisms: 1) *Zeroth-Order (ZO) Optimization*. By eliminating the need to transmit backward gradients, DPZV mitigates conventional gradient-based leakage (He et al., 2019). Specifically, no unperturbed gradients are ever exchanged among participants. 2) *Controllable Differential Privacy (DP)*. Each participant interacts only through black-box queries and responses augmented with DP-based noise. This randomized protocol further shields sensitive information from reconstruction or inference attacks.

## F.1    PROTECTION AGAINST AN HONEST-BUT-CURIOUS THREAT MODEL

In an *honest-but-curious* scenario, adversaries seek private insights into labels without deviating from the protocol. In this scenario, the most common attack is the Label Inference Attack. Such attacks typically exploit exact gradient signs or rely on known model architectures to backtrack label information (Fu et al., 2022; Li et al., 2021). Under DPZV, *unbiased gradients are never shared*, and the server model remains inaccessible to other participants. Clients observe only *stochastic ZO outputs* with added noise, preventing them from reverse-engineering labels via gradient signals.

Moreover, label inference methods often assume knowledge of the dataset domain or direct model access, which is not given in DPZV's design. Task details (e.g., label distributions, model layers) are held privately by each party, limiting the adversary's capacity to launch sophisticated inversion or inference attacks.

## F.2    PROTECTION AGAINST POST-TRAINING ATTACKS

Post-training adversaries typically aim to infer sensitive data from a *final, trained model* (Ateniese et al., 2015). Since DPZV applies differential privacy *throughout* training, the final model parameters satisfy rigorous $(\epsilon, \delta)$-DP guarantees. Hence, even if the trained model is released or accessed, the level of noise injected ensures that the adversary cannot reliably distinguish any single individual's data—reducing vulnerability to membership inference or model-inversion attacks (Jiang et al., 2022). This formal DP framework remains robust regardless of downstream usage or queries on the finalized model.

In summary, by combining ZO optimization with DP noise injection, DPZV provides comprehensive protection against both *honest-but-curious* and *post-training* adversaries across diverse privacy threat scenarios.

# G    MEMORY COST

Figure 5 compares the GPU memory consumption, with memory values normalized for readability. We compare the memory cost on larger models, where we use ResNet for image classification and

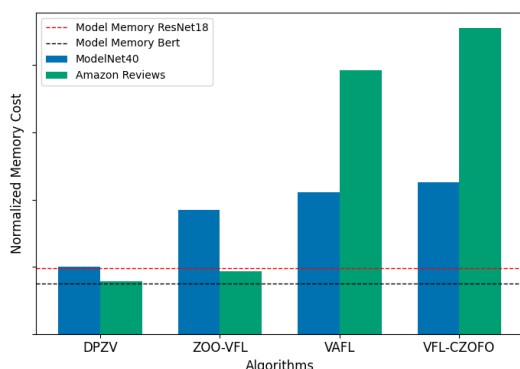

Figure 5: Normalized memory cost in training for each method. `DPZV` requires the smallest memory allocation in both datasets, almost the same as model memory itself. This shows the memory efficiency of `DPZV`, allowing superior performance on large-scale neural networks.

BERT for sequence classification. We record the highest memory peak to show the total required memory for each method on training large models. We observe that `DPZV` requires memory approximately equal to the model size, whereas the first-order method `VAFL` demands more than twice the model size. Compared to `ZOO-VFL`, `DPZV` achieves further memory savings by leveraging `MeZO`. *These results highlight the scalability of `DPZV`, making it well-suited for deploying large pretrained language models in VFL scenarios.*

## H  CHOICE OF ALL HYPERPARAMETERS

In this section, we show all the chosen Hyperparameters, including learning rate for the four datasets and ZO and DP parameters in Sec. 6:

Table 2: All learning rate choices

| DATASET | | VAFL | ZOO-VFL | VFL-CZOFO | DPZV (OURS) |
|---|---|---|---|---|---|
| MNIST | $\eta_0$ | 0.005 | 0.005 | 0.005 | 0.005 |
| | $\eta_m$ | 0.001 | $5 \times 10^{-5}$ | $1 \times 10^{-7}$ | $5 \times 10^{-4}$ |
| CIFAR-10 | $\eta_0$ | 0.005 | 0.005 | 0.005 | 0.005 |
| | $\eta_m$ | 0.001 | $5 \times 10^{-5}$ | $5 \times 10^{-8}$ | $1 \times 10^{-4}$ |
| MODELNET40 | $\eta_0$ | $1 \times 10^{-4}$ | $1 \times 10^{-5}$ | $5 \times 10^{-7}$ | $1 \times 10^{-5}$ |
| | $\eta_m$ | $1 \times 10^{-5}$ | $1 \times 10^{-5}$ | $5 \times 10^{-7}$ | $1 \times 10^{-5}$ |
| AMAZON REVIEW | $\eta_0$ | $1 \times 10^{-5}$ | $5 \times 10^{-7}$ | $1 \times 10^{-6}$ | $5 \times 10^{-7}$ |
| | $\eta_m$ | $1 \times 10^{-5}$ | $5 \times 10^{-7}$ | $1 \times 10^{-6}$ | $5 \times 10^{-7}$ |

Table 3: Other Hyperparameters

| PARAMETER | VALUE | EXPLANATION |
|---|---|---|
| $C$ | 10 | *DP clipping threshold* |
| $\lambda$ | 0.001 | *Scale of perturbation for ZOO* |
| $m$ | 0.9 | *Momentum parameter* |
| $\delta$ | $1 \times 10^{-5}$ | *DP error level* |
| $n$ | 5 | *Number of perturbation for VFL-CZOFO* |

## I   LLM USAGE

An LLM was used solely for grammar refinement and stylistic editing. No part of the technical content, experimental design, or analysis was generated by the model.

