# OpenReview forum: "Elevating the Tradeoff between Privacy and Utility in Zeroth-Order Vertical Federated Learning"
_ICLR.cc/2026/Conference — ICLR 2026 Conference Withdrawn Submission_

### Official Review · Reviewer_WTGp · 2025-10-26

**Soundness:** 3
**Presentation:** 1
**Contribution:** 2
**Rating:** 4
**Confidence:** 4

**Summary:**

The authors propose DPZV, the first tunable DP framework built upon ZOO-VFL. The framework extends the existing ZOO-VFL structure by introducing scalar-valued DP noise injection in the backward communication from the server to the clients, enabling controllable privacy levels while maintaining model utility.

**Strengths:**

1) The paper proposes a controllable differential privacy mechanism within ZOO-VFL. This design effectively integrates privacy control into ZO-based training, offering an adjustable trade-off between privacy and utility while maintaining convergence guarantees.

**Weaknesses:**

There are presentation and experiment design problems.

1) #140 regarding MeZO: Your paper did not specify how you use MeZO. Which mechanism of MeZO is applied? And why does it help reduce memory cost in your framework? This part of your paper is unclear; you should specify it in the Method section. Besides Figure 5, the experiment is missing context.

2) #248 confusing scheme: You mention that your server updates its model using (i) ZO optimization and (ii) SGD. However, in Algorithm 1, you state that the server is updated via ZO optimization. Moreover, in your experiments, you did not specify which optimization method was used. These inconsistencies make the method confusing.

3) Unfair baseline penalization: You add the DP mechanism by clipping the embeddings and adding calibrated vector noise to VFL-CZOFO and ZOO-VFL. Those methods are designed for VAFL only and are not suitable for these ZO-based methods, especially given that your approach adds scalar DP noise injection based on ZOO-VFL. Therefore, it is natural that those methods perform poorly under these circumstances.

4) Quasonabel choice of parameter: Regarding your choice of parameters in Table 2 (#1384), since your method applies scalar DP noise injection based on ZOO-VFL, I wonder why there is a significant difference in the order of magnitude of the learning rate $\eta_m$ between your method and ZOO-VFL or VFL-CZOFO in.


Minor:
1) The figure at the top of page 7 is missing its label. It should be marked as Figure 2
2) #333 and #371). Inconsistently uses “ZOFO” and “VFL-CZOFO” to refer to the same baseline. Use the same one.

**Questions:**

See weaknesses.

---

### Official Review · Reviewer_Aghn · 2025-10-27

**Soundness:** 2
**Presentation:** 3
**Contribution:** 2
**Rating:** 2
**Confidence:** 4

**Summary:**

The paper proposed DPZV, a vertical federated learning framework with DP protection, optimized via a Zero-Order approximation to protect label privacy. DPZV leverages ZO optimization to reduce communication overhead, since the gradient sent back to the client is reduced from a vector to a scalar. A DP guarantee protects the sent back gradient. Extensive experimental and theoretical analyses are conducted to highlight the advantages of the proposed method.

**Strengths:**

- The proposed method incorporates an existing mechanism that is easy to follow and re-implement.
- Extensive theoretical analysis and experimental results are conducted to highlight the advantages.
- The paper is well-written and easy to follow.

**Weaknesses:**

- Vague threat model. I suggest that the author provide a precise analysis of the threat model: who they are and what their goals are. And what are they capable of? The key reason is that although the proposed method incorporates a Gaussian mechanism for DP protection, this mechanism only provides membership-level protection, leaving out other risks, such as label reconstruction attacks, as discussed in the paper's motivation.
- The motivation for using ZO optimization is incorrect. The paper claims that it uses ZO-optimization to reduce communication overhead by reducing the gradient from a vector (or matrix) to a scalar for the loss function. However, to compute the ZO approximation, you have to send two vectors (or two matrices) to compute the approximation, which is the exact communication requirement (even one more for the scalar). Thus, the motivation for this is not convincing to me.
- The experimental results consider unrelated baselines. Specifically, the goal of DPZV is to protect the label held by the server. In contrast, its consideration of baselines (such as VAFL) protects the privacy of the embeddings extracted from clients. This mismatch reduces the validity of the claims concluded from the experimental results.

**Questions:**

- In lines 190-191, why can the server hold $\xi_i\in D$? For VFL, the server only holds the label of $\xi_i$.
- How did the author configure the parameter $\mu$ to achieve the desired privacy budget $\epsilon$ and broken probability $\delta$?
- Corollary 5.2., why $h$ is DP w.r.t the label? Should it be the gradient w.r.t $h$?

---

### Official Review · Reviewer_qstq · 2025-10-31

**Soundness:** 2
**Presentation:** 3
**Contribution:** 2
**Rating:** 2
**Confidence:** 4

**Summary:**

The paper proposed a differentially private VFL algorithm that aims to protect the potential leakage during the backward gradient communication against honest-but-curious clients. The proposed algorithm uses a zero-order (ZO) mechanism to avoid sharing gradients during the client-server communication. Differential Privacy (DP) noise is added by the server before broadcasting the global updates to clients.

**Strengths:**

- The paper discusses rigorous convergence and privacy guarantee of the proposed method.
- Overall, the presentation is clear and easy to follow.

**Weaknesses:**

- It seems like there is no privacy guarantee in forward communication as clients simply perform the first step of ZO computation. The author mentioned that ZO does not prevent malicious clients from approximating gradients from perturbed information (line 50). If the server is malicious, this is a serious privacy leak that is not addressed by the algorithm.
- Related to the above, the paper does not make the attack model clear. Who are they trying to protect information from? And what information is begin protected? The latter needs to be clarified in Theorem 5.1.
- Experiment baselines have DP guarantees in forward communication that would protect shared data against the server, while the proposed method does not have this protection. The privacy guarantee and threat models are different between the proposed method and the baselines; hence, it is not a fair comparison.
- The paper mentions the benefit of reduced communication via the ZO method. It would be nice if the paper can show empirically how the proposed method saves on communication cost compared to other methods.
- Novelty is limited. Zero-order VFL is not new, and the main contribution is from scalar noise injection which is applied in the backward instead of the forward communication. However, as pointed out above, this does not provide the same privacy guarantees.

**Questions:**

- How does the server update its parameters? For a first gradient, the server needs the unperturbed embeddings, which it does not have. For a ZO gradient, can you provide the update step details?
- What is the role of $\mu$ in Theorem 5.1? It seems to be the main privacy parameter, but it is not used anywhere else (e.g. Empirical studies use $\epsilon$). I think Theorem 5.1 can be done with just the standard privacy parameters epsilon and delta, so $\mu$ is a bit redundant.

---

### Official Review · Reviewer_v1qv · 2025-10-31

**Soundness:** 2
**Presentation:** 3
**Contribution:** 2
**Rating:** 4
**Confidence:** 3

**Summary:**

The paper proposes DPZV, a differentially private zeroth-order VFL algorithm for nonconvex optimization, providing convergence, privacy, and utility guarantees under standard stochastic optimization assumptions, including Lipschitz continuity, smoothness, variance-bounded gradients, bounded communication delays, and partial client participation. Extensive experiments on four benchmark datasets show that DPZV outperforms existing zeroth-order VFL methods, especially under strict privacy constraints $\epsilon \leq 1$.

**Strengths:**

- The convergence, differential privacy, and utility guarantees of the proposed algorithm called DPZV are derived for nonconvex optimization problems. In particular, the analysis relies on standard assumptions for stochastic optimization, including: (1) Lipschitz continuity and smoothness of the objective functions, (2) variance-bounded stochastic gradients, (3) upper-bound on communication delays, and (4) probability of a client participating in each communication round.



- Extensive empirical evaluation compares DPZV with existing zeroth-order VFL algorithms on four benchmark datasets. In particular, DPZV outperforms baselines across different settings (e.g., private and non-private), particularly under strict privacy constraints $\epsilon \leq 1$.

**Weaknesses:**

- DPZV is very similar to DPGD with zeroth-order gradients by Zhang et al. (2024), with the main distinction being that DPZV uses scalar-valued noise for privacy guarantees and supports an asynchronous protocol.

- Although asynchrony delays are assumed to be bounded, Theorem 4.4 does not account for their effect on the step-size or convergence. This is unusual, as asynchronous or delayed gradient methods, e.g., [1,2], typically adjust the stepsize to account for delays, which directly influences convergence bounds, as also shown by Zhang et al. (2021) for their proposed zeroth-order algorithms.

- According to Theorem 4.4, the clipping level (C) must be sufficiently large; otherwise, the clipping operator could effectively be inactive.

[1] Feyzmahdavian, H. R., & Johansson, M. (2023). Asynchronous iterations in optimization: New sequence results and sharper algorithmic guarantees. _Journal of Machine Learning Research, 24_(158), 1–75.

[2] Gurbuzbalaban, M., Ozdaglar, A., & Parrilo, P. A. (2017). On the convergence rate of incremental aggregated gradient algorithms. _SIAM Journal on Optimization, 27_(2), 1035–1048.

**Questions:**

- The authors could expand on the use of scalar-valued DP noise and clarify its relation to Equation (8), which appears to involve vector-wise DP noise.

- Local embeddings are not defined.

- Is it necessary for stochastic gradient noises to be unbiased? Assumption 4.2 does not seem to require the unbiasedness of stochastic gradient noises.

- $B$ and $\gamma_1$ in Theorem 4.4 are not defined.

- $D$ and $\mu$ in the last paragraph of Section 4 (right before Section 5) are not defined.

- Does the DP definition refer to client-level differential privacy?

---

### Note · Authors · 2025-11-12

I have read and agree with the venue's withdrawal policy on behalf of myself and my co-authors.